https://doi.org/10.1038/s41467-018-07864-w　　**OPEN**

# Structure of a human intramembrane ceramidase explains enzymatic dysfunction found in leukodystrophy

Ieva Vasiliauskaité-Brooks [1], Robert D. Healey [1], Pascal Rochaix[1], Julie Saint-Paul[1], Rémy Sounier[1], Claire Grison[1], Thierry Waltrich-Augusto[1], Mathieu Fortier[1], François Hoh[2], Essa M. Saied [3,4], Christoph Arenz[3], Shibom Basu[5], Cédric Leyrat [1] & Sébastien Granier[1]

Alkaline ceramidases (ACERs) are a class of poorly understood transmembrane enzymes controlling the homeostasis of ceramides. They are implicated in human pathophysiology, including progressive leukodystrophy, colon cancer as well as acute myeloid leukemia. We report here the crystal structure of the human ACER type 3 (ACER3). Together with computational studies, the structure reveals that ACER3 is an intramembrane enzyme with a seven transmembrane domain architecture and a catalytic $Zn^{2+}$ binding site in its core, similar to adiponectin receptors. Interestingly, we uncover a $Ca^{2+}$ binding site physically and functionally connected to the $Zn^{2+}$ providing a structural explanation for the known regulatory role of $Ca^{2+}$ on ACER3 enzymatic activity and for the loss of function in E33G-ACER3 mutant found in leukodystrophic patients.

[1] IGF, University of Montpellier, CNRS, INSERM, Montpellier 34094, France. [2] CBS, University of Montpellier, CNRS, INSERM, Montpellier 34090, France. [3] Institute for chemistry, Humboldt-Universität zu Berlin, Brook-Taylor-Str. 2, 12489 Berlin, Germany. [4] Chemistry Department, Faculty of Science, Suez Canal University, 41522 Ismailia, Egypt. [5] Macromolecular Crystallography, Swiss Light Source, Paul Scherrer Institut, 5232 Villigen PSI, Switzerland. These authors contributed equally: Ieva Vasiliauskaité-Brooks, Robert D. Healey. Correspondence and requests for materials should be addressed to C.L. (email: cedric.leyrat@igf.cnrs.fr) or to S.G. (email: sebastien.granier@igf.cnrs.fr)

The main bioactive sphingolipids ceramide, sphingosine, and sphingosine 1-phosphate (S1P) play key roles in human (patho)physiology including cancer cell biology, immune, inflammatory, and metabolic functions (reviewed in ref. [1]). As a result, enzymes regulating sphingolipid levels constitute key therapeutic targets, particularly for the treatment of cancer[2]. Among these enzymes, ceramidases (CDases) are attractive targets for clinical intervention[3] as they directly regulate the balance between these bioactive lipids by converting ceramides into free fatty acids and sphingosine[4] which is further processed into S1P by kinases[5].

The five ceramidases cloned to date are classified into acid, neutral, and alkaline groups according to the pH optima of the hydrolysis reaction (reviewed in ref. [3]). However, the three groups do not display any sequence homology; the acid ceramidase (ASAH1), ubiquitously expressed, is mainly present in lysosomes, its inactivation by mutation causing Farber disease[6]. The recent crystal structures of ASAH1 revealed a globular fold associating α-helices and anti-parallel β-sheets[7]. This study also showed that the ASAH1 enzymatic activity necessitates an autoproteolytic-based conformational change exposing the putative substrate binding cavity and the cysteine-based catalytic center at its base[7]. The neutral ceramidase (NCDase) is also ubiquitously expressed, structurally containing one transmembrane domain (TM) and a large soluble domain[8] unrelated to ASAH1. The recent crystal structure of NCDase soluble domain revealed a $Zn^{2+}$-dependent catalytic site deeply buried in a hydrophobic binding pocket which can accommodate the ceramide[9].

Alkaline ceramidases (ACERs) are much less well-understood, in part because of their hydrophobic nature that, until now, has rendered the biochemical and structural analyses difficult. Three different genes have been cloned—ACER1[10], ACER2[11], and ACER3[12], and sequence analyses suggest that they are integral membrane proteins. ACER1 and ACER2 expression is rather tissue specific (skin and placenta, respectively), while ACER3 is expressed in most tissues[10–12]. Very little is known at the molecular level: ACERs are localized intracellularly in the membrane of the endoplasmic reticulum-Golgi apparatus network and their activity, mainly directed against ceramides with long unsaturated acyl chains (C18:1, C20:1, and C24:1), was shown to be $Ca^{2+}$-dependent[10,12–14].

The critical role of ACERs in human physiology and, in particular ACER3, was recently revealed by clinical data demonstrating that ACER3 deficiency leads to progressive leukodystrophy in early childhood[15], a disease for which no treatment is available today. This study demonstrated that patients were homozygous for a p.E33G ACER3 mutant and that this mutation impaired the ACER3 ceramidases activity in patients' cells. When compared to healthy individuals, this loss of function resulted in higher level of several ceramide species in the blood, in particular for the ACER3 preferred substrates, C18:1 and C20:1 ceramides. It was proposed that these aberrant levels of ceramides in the brain could result in an incorrect central myelination leading to the clinical phenotype associated with the ACER3 mutant, i.e., neurological regression at 6–13 months of age, truncal hypotonia, appendicular spasticity, dystonia, optic disc pallor, peripheral neuropathy, and neurogenic bladder[15]. However, in mice, while ACER3 knock-out results in an aberrant accumulation of various ceramides, it does not affect myelination. Instead, this deficiency induces the premature degeneration of Purkinje cells and cerebellar ataxia[16]. In the periphery, in mice, the modulation of C18:1 ceramide levels by ACER3 regulates the immune response through the upregulation of cytokines, while its deficiency increases colon inflammation and its associated tumorigenesis[17]. Moreover, in vitro results obtained in human cells revealed that ACER3 contributes to acute myeloid leukemia (AML) pathogenesis[18]. Indeed, it was found that ACER3 expression negatively correlates with the survival of AML patients, and that ACER3 is essential for the growth of AML cells as the sh-RNA inhibition of its expression resulted in an increase of apoptosis and in an important decrease in cell growth[18].

Modulating ceramide homeostasis can have broad implications and targeting ACER3 for clinical purposes will be extremely challenging. More research is needed to determine whether the molecular control of ACER enzymatic activity (agonists and antagonists) could constitute a possible clinical intervention for the treatment of leukodystrophy, colon cancer, or AML among other pathologies involving ceramide dyshomeostasis. The first step toward this endeavor is to better understand the molecular basis of ACER function.

In an effort to determine the molecular mechanisms of the ACER enzymatic function, we undertook a structural analysis of ACER3. We solve a 2.7 Å crystal structure using in meso crystallography that, together with computational studies, reveal at the atomic level how this integral membrane enzyme accommodates and hydrolyzes its ceramide substrate in a $Zn^{2+}$- and $Ca^{2+}$-dependent manner. Furthermore, we uncover how a single point mutant (E33G) results in the destabilization of the calcium binding site, providing a molecular explanation for the ACER3 mutant dysfunction leading to leukodystrophy in human.

## Results

**ACER3 is a seven transmembrane protein.** Recent breakthroughs in the crystallization of challenging membrane proteins such as G protein-coupled receptors include the use of protein engineering with a soluble module, an approach pioneered in 2007 to solve the structure of the beta2-adrenergic receptor[19]. Here, we modified ACER3 (residues 1–244) with the BRIL soluble module (thermostabilized apocytochrome b562RIL from *Escherichia coli*[20]) fused to its C-terminus (Supplementary Figure 1 and Methods) and showed that this fusion is biologically active; it hydrolyzes C18:1 ceramide substrate in a preferred manner over C18:0 ceramide in detergent micelles (Supplementary Figure 1). The ACER3–BRIL fusion was crystallized in a cholesterol-doped monoolein lipidic mesophase and the structure determined at 2.7 Å resolution (Table 1).

When cloned in 2001, ACER3 was predicted to be an integral membrane protein with five TMs[12]. In fact, the crystal structure revealed that it possesses seven alpha-helices forming a 7TM architecture with opposite N- and C-terminus domains (Fig. 1). Strikingly, the fold of ACER3 is similar to adiponectin receptors (ADIPORs) with the 7TM harboring a $Zn^{2+}$ binding site[21] (Fig. 1a, Supplementary Figure 2) despite a very low sequence identity (14% with ADIPOR1 and 10% for ADIPOR2). This similar 7TM fold suggests that ACER3 and ADIPORs possess the same topology; an intracellular N-terminus exposed to the cytoplasm and the C-terminus facing the lumen (Fig. 1, Supplementary Figure 2). We confirmed this topology in living cells using time-resolved fluorescence resonance energy transfer (TR-FRET) experiments (Supplementary Figure 2). On each side, the TMs are connected by three cytoplasmic loops (ICL 1–3) and three extracellular loops (ECL 1–3) (Fig. 1a). ICL3, connecting the TM6 and 7, is the most notable loop (residues 196–216) as it forms a motif composed of two short alpha-helices (one and two turns) lying parallel to the membrane plane connected by a positively charged loop (a RKKVPP sequence) that is suitably positioned to interact with the polar head of negatively charged lipids (Fig. 1b).

The N-terminus (1–34), devoid of secondary structure, is packed against the bottom of the TM bundle (1–3, 6, and 7) and forms a spiral-like motif in which resides a $Ca^{2+}$ ion.

**Table 1 Data collection and refinement statistics (molecular replacement)**

|  | ACER3–BRIL, native[a] | ACER3–BRIL, Zn edge |
|---|---|---|
| *Data collection* |  |  |
| Space group | C222₁ | C222₁ |
| *Cell dimensions* |  |  |
| a, b, c (Å) | 60.88, 68.83, 257.52 | 61.41, 69.69, 258.15 |
| α, β, γ (°) | 90.00, 90.00, 90.00 | 90.00, 90.00, 90.00 |
| Resolution (Å) | 45.60–2.70 | 46.07–2.85 |
|  | (2.83–2.70) [b] | (3.00–2.85) |
| $R_{merge}$[c] | 0.566 (NA) | 0.614 (NA) |
| $R_{meas}$ | 0.577 (NA) | 0.619 (NA) |
| $R_{pim}$ | 0.110 (0.875) | 0.102 (NA) |
| $CC_{1/2}$ | 0.964 (0.496) | 0.999 (0.349) |
| $I/\sigma I$ | 8.1 (1.1) | 7.8 (0.7) |
| Completeness (%) | 100.0 (100.0) | 100.0 (100.0) |
| Redundancy | 27.6 (27.7) | 68.8 (65.2) |
| Wilson B factors | 73.6 | 62.7 |
| *Refinement* |  |  |
| Resolution (Å) | 45.60–2.70 | – |
| No. of reflections | 15,331 | – |
| $R_{work}/R_{free}$ | 24.90/27.14 | – |
| *No. of atoms* |  |  |
| Protein | 2855 | – |
| $Zn^{2+}$ | 1 | – |
| $Ca^{2+}$ | 1 | – |
| $Mg^{2+}$ | 2 |  |
| $Na^+$ | 3 |  |
| $SO_4^{2-}$ | 30 |  |
| Mono-olein | 25 | – |
| Water | 122 | – |
| *B-factors (Å²)* |  |  |
| Protein | 74.63 | – |
| $Zn^{2+}$ | 86.05 | – |
| $Ca^{2+}$ | 97.43 | – |
| $Mg^{2+}$ | 50.97 |  |
| $Na^+$ | 75.02 |  |
| $SO_4^{2-}$ | 113.02 |  |
| Mono-olein | 87.20 | – |
| Water | 57.08 | – |
| *R.M.S. deviations* |  |  |
| Bond lengths (Å) | 0.010 | – |
| Bond angles (°) | 0.969 | – |

[a]The reported statistics correspond to datasets that were obtained by merging serial data from many different crystals collected in wedges of 10° (77 and 198 wedges for the native Zn edge datasets, respectively)
[b]Values in parentheses are for highest-resolution shell
[c]In this case, indicators like Rmerge and Rmeas are not suitable for assessing data quality[37]
NA not applicable, $R_{merge}$ value over 1 is statistically meaningless

The crystal structure uncovers a disulfide bond formed between C21 and C196 connecting the N-terminus to TM6. This disulfide may play an important role in stabilizing this peculiar N-terminal domain fold (Fig. 1c). In a cellular context, this disulfide is facing the reducing environment of the cytoplasm and we cannot exclude the possibility that it is dynamically regulated by its redox local environment, in particular neighboring charged residues/lipids may lower thiolate pKa and render those more prone to oxidation[22].

**Intramembrane binding sites**. The ACER3 7TM domain contains a large hook-shaped intramembrane pocket directly accessible to the lipid leaflet through two openings between the TM5 and the TM6 (Fig. 2). This cavity is formed by residues belonging to all TMs except TM4 (Fig. 2a). At the top, the V164$^{TM5}$ side chain is capping the pocket, while V157$^{TM5}$, I161$^{TM5}$, Y176$^{TM6}$, and L179$^{TM6}$ side chains shape the side of the cavity facing the

lipids (Fig. 2b, c). Opposed to this hydrophobic patch and toward the inner core of the TM, there is a rather polar environment formed by the residues S160$^{TM5}$, S178$^{TM6}$, and H231$^{TM7}$ in which we found some electron density that we modeled as a water molecule (Fig. 2b). Such a hydrophilic patch might play a role in ceramide binding selectivity, as it may prevent the binding of ceramide harboring acyl chains over 24 carbons. Directly below this area appeared a large electron density in the calculated 2Fo–Fc map as well as in the (unbiased) polder OMIT map[23] (see Methods) (Supplementary Figure 3) that we tentatively modeled as a mono-olein. Indeed, mono-olein by weight forms 54% of the lipidic cubic phase (i.e., ~1.9 M) and displays an acyl chain moiety chemically identical to that of C18:1 ceramide (Supplementary Figure 3), the preferred substrate of ACER3[14].

At the bottom of the intramembrane pocket, we unambiguously identified a $Zn^{2+}$ ion (Supplementary Figure 3 and Methods). This $Zn^{2+}$ is coordinated by three His residues (H81$^{TM2}$, H217$^{TM7}$, and H221$^{TM7}$) and a water molecule forming hydrogen bonds with S77$^{TM2}$ and D92$^{TM3}$ (Fig. 2d). The residues forming this $Zn^{2+}$ binding site are strictly conserved among the ACER family, and across species from yeast to human, highlighting the importance of this site in the biological function of these proteins (Supplementary Figure 4). Given the well-characterized ceramidase activity of ACER3, these structural data definitely establish ACERs as $Zn^{2+}$-dependent intramembrane enzymes.

**Substrate docking and hydrolysis mechanism**. We then assessed the mechanism of substrate binding and hydrolysis. To this end, we used computational docking and molecular dynamics (MD) simulations to calculate the most favorable binding mode of a C18:1 ceramide into the ACER3 lipid binding pocket (Fig. 3). The best scoring docking pose positions the ceramide in the hook-shaped cavity within the 7TM (Fig. 3a). The C18:1 acyl chain is stabilized through extensive contacts with hydrophobic residues lining the interior of the cavity including M43$^{TM1}$, V73$^{TM2}$, M96$^{TM3}$, F103$^{TM3}$, V153$^{TM5}$, L156$^{TM5}$, V157$^{TM5}$, and F182$^{TM6}$ (Fig. 3b). Surprisingly, the site of unsaturation with the sp$^2$ carbons is surrounded by a polar environment constituted by the sulfhydryl group of C100$^{TM3}$ and three hydroxyl groups of S99$^{TM3}$, Y149$^{TM5}$, and S228$^{TM7}$ (Fig. 3c). We functionally tested S99A, Y149A, and S228A mutants and compared their enzymatic activity with the one of ACER3–BRIL wild-type (WT) preparations. In agreement with the docking pose, the Y149A mutant presented an important decrease in activity, while S99A and S228A mutants did not show any significant functional differences (Supplementary Figure 5). Moreover, none of the mutants showed a change in the substrate preference (Supplementary Figure 5), suggesting that they alone are not critically involved in this selectivity. The sphingosine moiety remains partially accessible to the membrane leaflet and interacts with a set of hydrophobic residues from TM5 and TM6 (Fig. 3b). The ceramide carbonyl group and its primary alcohol directly interact with the $Zn^{2+}$ ion, resulting in an octahedral coordination sphere (Fig. 3d). The ceramide is further stabilized by polar contacts between its carbonyl and the amine group of W220$^{TM7}$ side chain, as well as its primary alcohol and the carboxylate of D92$^{TM3}$. Interestingly, the crystallographic water molecule that bridges S77$^{TM2}$, D92$^{TM3}$, and the $Zn^{2+}$ ion appears to be suitably placed for a nucleophilic attack on the ceramide carbonyl (Fig. 3d), suggesting a general acid–base catalytic mechanism in which D92$^{TM3}$ acts as a proton acceptor/donor (similar to e.g., zinc-dependent proteases)[24] (Supplementary Figure 5). From a substrate specificity point of view, it is clear from the structure and the docking results that substrates presenting large/bulky

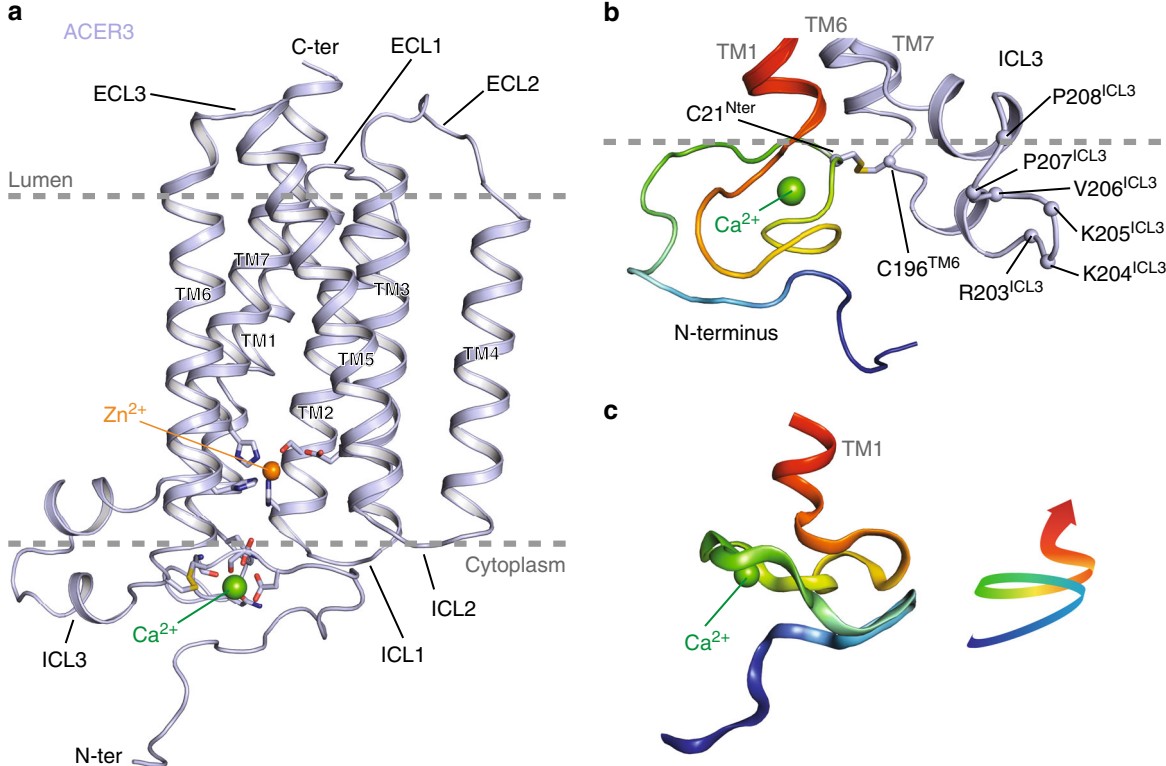

**Fig. 1** Crystal structure of ACER3. **a** Overall view of ACER3 crystal structure at 2.7 Å from within the membrane plane. The zinc and calcium ions are represented as spheres with a van der Waals radius of 0.88 and 1.14, respectively. Side chains of residues in close proximity to both ions are shown as sticks. **b** Close-up view of the N-terminus (colored from blue to red) and intracellular loop 3 (ICL3, colored in light blue) domains highlighting the disulfide bond formed by C21 and C196 (sticks). **c** Cartoon representation of the N-terminus domain colored as in (b) (left) revealing a spiral-like motif (right)

modifications of the primary alcohol such as sphingomyelin, glucosylceramide, or ceramide-1-phosphate cannot be accommodated in the pocket, and are unlikely to serve as ACER3 substrate (Supplementary Figure 5). Such a steric hindrance serving as the mechanism of substrate selectivity was also observed in neutral ceramidases[9].

Additional evidence supporting the biological relevance of the described ceramide binding pose was obtained from multiple simulations of the ACER3–C18:1 ceramide complex. Indeed, the initial binding pose was very stable with the all-atom root mean square deviations (RMSD) of the bound C18:1 ceramide remaining below 3 Å in every trajectory of five independent simulations (Fig. 3a and Supplementary Figure 5).

**Comparison to ADIPORs structures.** The overall architecture of ACER3 is similar to the ADIPORs. In particular, the position of the TM1–3, 6, and 7 forms a similar pattern when viewed from the cytoplasm or from the lumen (Fig. 4a). In addition to the similar 7TM architecture, the $Zn^{2+}$ binding sites with the canonical $(H)_3SD$-water module are almost superimposable (Fig. 4b). Of interest, the sequence $(H)_3SD$ is unifying a superfamily of putative hydrolases recently identified based on statistically significant sequence similarities, termed CREST (ACER, progesterone adipoQ receptor (PAQR) receptor, Per1, SID-1, and TMEM8)[25].

Significant/major differences are observed however between ACER3 and ADIPORs. Unlike ADIPORs, the ACER3 intramembrane pocket is not connected to the upper leaflet of the membrane nor to the cytoplasmic domain, due to a distinct N-terminus fold (Supplementary Figure 6). Conformational changes within domains accessible to the hydrophilic cytoplasm must occur for water molecules to access the active site. The difference

between the intramembrane pockets also resulted in distinct calculated ceramide binding modes differing essentially at the level of the sphingosine moiety position (Supplementary Figure 6). These differences may have some impact on the intrinsic enzymatic properties of ACER3 vs. ADIPORs, i.e., $K_M$ parameters and substrate preference.

A key difference between ACER3, ADIPOR1, and ADIPOR2 is seen in the TM4 and TM5 architecture. First, the ACER3 TM4 is much shorter than that of the ADIPORs, resulting from a longer ECL2 connecting TM3 and TM4 (Fig. 4c). Second, as compared to the open ADIPOR1 and closed ADIPOR2 TM4 and TM5 structures, ACER3 presents an intermediate TM4-TM5 conformation (Fig. 4c). These architectural differences are most likely originating from the biochemical preparations as well as from crystallization conditions: where a closed ADIPOR2 structure favored the binding of an oleic acid; an intermediate ACER3 bound a monoolein; and an open ADIPOR1 structure was ligand-free. The three distinct conformations might however represent distinct steps of the common catalytic process, the conformational changes and dynamics of such action of intramembrane ceramidases remains to be explored.

On the other hand, the described structure of the soluble NCDase has nothing in common with the ACER3 structure apart from the three histidines coordinating the $Zn^{2+}$ (Supplementary Figure 6). In particular, catalytic sites belong to two clearly distinct scaffolds: intramembranous for ACER3 and membrane-associated for NCDase with a water-soluble domain predominantly constituted of beta-sheets (Supplementary Figure 6) forming the so-called beta-triangle hydrolase scaffold[9]. These clearly distinct structures are in agreement with the different cellular localization as well as with the functional specialization of these enzymes[3].

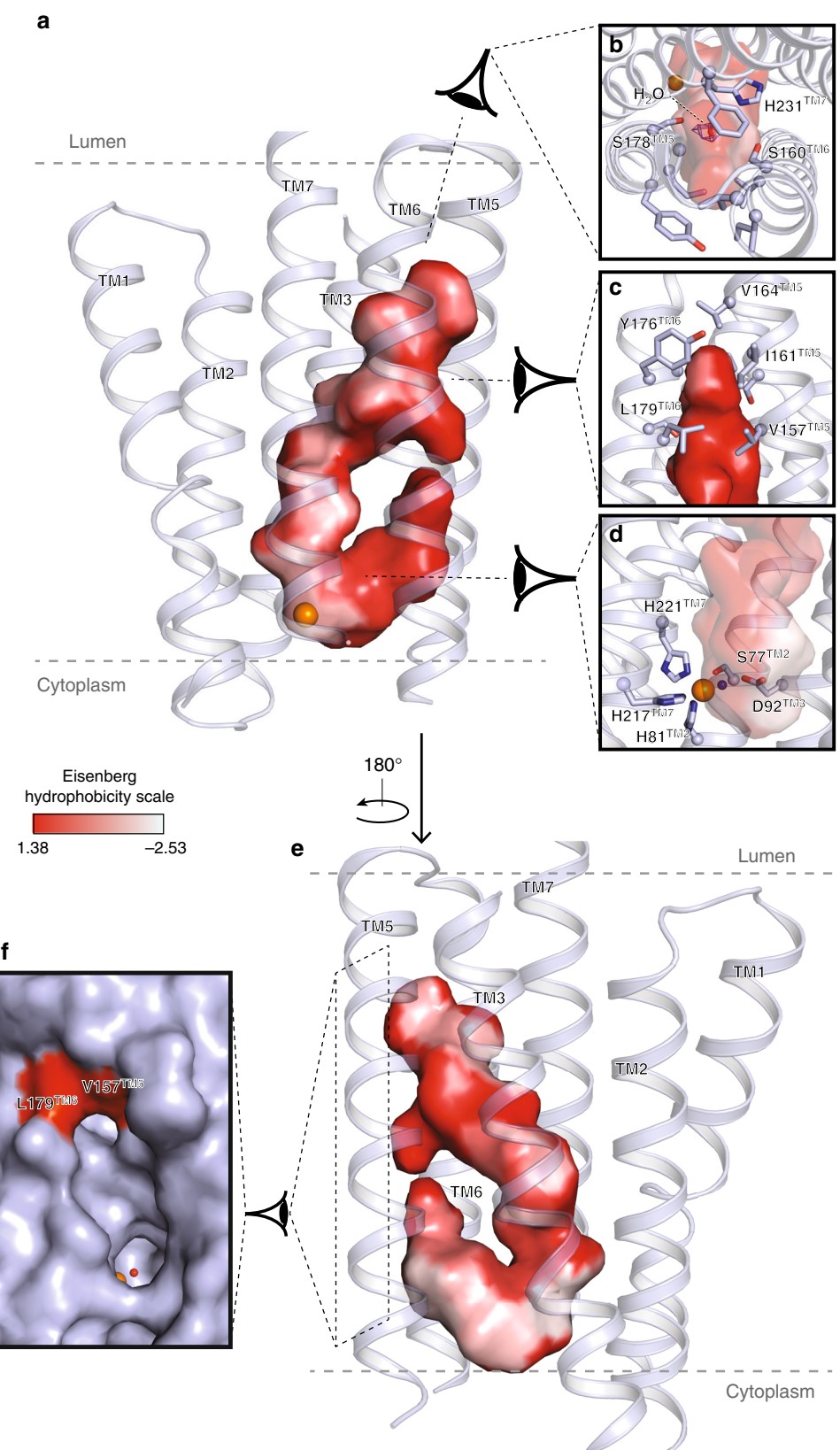

**A Ca²⁺-binding site within the N-terminus**. Perhaps the most interesting feature of the ACER3 structure is the presence of a $Ca^{2+}$-binding site within the N-terminal domain (Fig. 5a). $Ca^{2+}$ was shown to modify the enzymatic activity of ACER3[10,12–14], but the molecular basis for this effect was unknown. The $Ca^{2+}$ ion is coordinated by six oxygens from the D19 carboxylate (bidentate), the W20 backbone carbonyl, the E22 backbone carbonyl, the N24 side chain carbonyl, and the E33 carboxylate (monodentate) (Fig. 5b). The resulting coordination geometry resembles an incomplete pentagonal bipyramid. However, during

**Fig. 2** ACER3 intramembrane domain. **a** View of the large hook-shaped internal cavity shown as surface (cavity mode 1) within the 7TM helix bundle (shown in light blue cartoon). The TM4 has been removed for clarity. The cavity is colored according to the Eisenberg hydrophobicity classification from red (high hydrophobicity) to white (low hydrophobicity). **b**, **c** Close-up views of the pocket on the top highlighting the observed density (blue mesh, 2Fo–Fc map contoured at $1\sigma$) in which a water molecule was modeled (red sphere) **b** and on the side **c** with residues lining the pocket shown as sticks. **d** Close-up view of the zinc binding site highlighting the residues forming the first coordination sphere of the $Zn^{2+}$ shown as sticks. The modeled water molecule is shown as a blue sphere. **e** 180° rotation of the view described in (**a**). **f** Side view of the pocket shown as surface (cavity mode 0) revealing the pocket accessibility at the level of the $Zn^{2+}$ site and right above it

MD simulations, the coordination of D19 carboxylate switches from bidentate to monodentate, and the coordination sphere is completed by a water molecule resulting in an octahedral calcium site (Supplementary Figure 7). The loop forming the $Ca^{2+}$ binding site is connected to the 7TM core through a disulfide bond formed by C21 and $C196^{TM6}$ positioned at the bottom of TM6 (beginning of the ICL3). The sequences $D_{19}WCE(X)N_{24}$ and $_{32}AEF_{34}$ constituting the $Ca^{2+}$ binding domain as well as the $C196^{TM6}$ forming the disulfide bond are strictly conserved among the ACER family (Fig. 5c) in agreement with the $Ca^{2+}$-dependent ceramidase activity described for ACER1, ACER2, and ACER3[10,12–14]. Moreover, this domain is also strictly conserved across species from yeast to human (Fig. 5c, Supplementary Figure 4). This strict conservation during evolution further highlights the paramount regulatory role of $Ca^{2+}$ in the enzymatic function.

Interestingly, based on the crystal structure and MD simulations results, two residues appear to play critical roles in physically linking the $Ca^{2+}$ and the $Zn^{2+}$ sites, W20 and E33. Indeed, in the structure, the two residues interact simultaneously with both metal sites (Fig. 5d). First, E33 carboxylate interacts with the $Ca^{2+}$ ion and is in close proximity to $H81^{TM2}$ side chain, while $H81^{TM2}$ coordinates the $Zn^{2+}$ (Fig. 5d). Interestingly, E33 carboxylate rearranges during MD simulations to form a hydrogen bond with $H81^{TM2}$ side chain (Fig. 5d, e). Second, W20 coordinates $Ca^{2+}$ through its backbone carbonyl, and its indole ring forms a His–aromatic complex with $H81^{TM2}$ and $H217^{TM7}$ side chains (Fig. 5d). Such His–aromatic interactions are found in other enzymes and constitute key components of the enzymatic catalytic mechanism by participating in the transition-state stabilization or through critical constraints on the conformation of the catalytic site (described in ref. [26]). In addition, W20 forms a network of hydrophobic interactions with L18, $F80^{TM2}$, $W189^{TM6}$ forming the bottom of the hook-shaped intramembrane pocket toward the cytoplasm (Fig. 5f). We hypothesize that the functional effect of $Ca^{2+}$ on the catalytic activity of ACER3 originates from this direct link, with changes to the $Ca^{2+}$ site propagating to the $Zn^{2+}$ catalytic site.

Remarkably, this discovery provides molecular insights into the E33G ACER3 mutation carried by patients suffering leukodystrophy, which results in the loss of ACER3 ceramidase activity despite similar level of expression than in control membrane preparations[15]. We used MD simulations to investigate the effect of this mutation on the stability of the ACER3 structure relative to the CER18:1-bound wild type model (Fig. 6). The effect observed on the flexibility of the $Zn^{2+}$ itself is rather mild due to the integrity of its coordinating residues and the presence of the stabilizing C18:1 ceramide molecule (Supplementary Figure 8). Of note, unlike the WT ACER3, the purified E33G ACER3 mutant was highly unstable during purification and yielded mostly aggregated protein eluting in the void volume during size exclusion chromatography, suggesting that the destabilization of the $Ca^{2+}$ site might influence the overall protein stability. In agreement with this experimental observation, MD simulations revealed rapid $Ca^{2+}$ unbinding followed by an overall destabilization of the N-terminal/cytoplasmic region of the protein in the mutated E33G model (Fig. 6, Supplementary Figure 8). At the

level of the $Ca^{2+}$ and $Zn^{2+}$ sites, the loss of $Ca^{2+}$ results in increased flexibility of W20 and motions of the $Ca^{2+}$ binding loop $(D_{19}WCE(X)N_{24})$ which deforms the substrate binding pocket and increases water accessibility to the $Zn^{2+}$ site (Supplementary Figure 8). These calculated alterations most likely represent a molecular explanation for the observed loss of E33G ACER3 ceramidase activity in patients' cells.

In order to further validate the role of the $Ca^{2+}$ binding site, we performed some additional enzymatic assays on ACER3 single point mutants D19G, E22G, N24G and E33G. As anticipated, we confirmed the functional data already published for the E33G mutant i.e., a dramatic decrease in the enzymatic function when compared to the WT preparations (Supplementary Figure 7). Two other mutants, E22G and N24G, behaved as E33G, displaying a clear decrease in enzymatic activities (Supplementary Figure 7). Surprisingly, the enzymatic activity of D19G was only partially affected (Supplementary Figure 7). Altogether, these data confirm the critical role of the $Ca^{2+}$ binding site in ACER3 function.

## Discussion

In this study, we solved the crystal structure of ACER3 revealing a seven-transmembrane domain architecture harboring a catalytic $Zn^{2+}$ binding site nearly identical to the one we recently uncovered in ADIPORs[21]. Considering the fact that ACER3 and ADIPORs share a similar fold, a common $Zn^{2+}$ catalytic site and similar functional enzymatic capability, it is highly likely that ADIPORs are indeed genuine ceramidases. These discoveries are expanding the family of ceramidases to, at least, seven members in humans. Given the established functional link between fungal PAQR and ceramidase activity[27], it is then tempting to speculate that all the members of the PAQR family[28] (11 proteins in humans including ADIPOR1 and ADIPOR2) share this functional characteristic with ACER3.

Moreover, our data provide a structural explanation of the previously demonstrated regulatory role of $Ca^{2+}$ on ACER3 enzymatic activity[12]. As it is known that high pH can increase the binding of $Ca^{2+}$[29], this might also constitute the molecular basis for the alkaline pH optima of ACER ceramidase activity in vitro[12].

Although ADIPORs seem to be ligand-dependent ceramidases (activated by adiponectin), the intracellularly residing ACER3 might be regulated through changes in cytoplasmic $Ca^{2+}$ concentration and/or local pH. In addition, the ACER3 structure and computational studies highlight the role of E33 in the formation of the $Ca^{2+}$ binding site and provide insights into the loss of ceramidase activity of the E33G–ACER3 mutant found in leukodystrophic patients. This knowledge will enable the development of pharmacological chaperones specifically designed to restore the enzymatic function of ACER3 mutant. This may constitute a potential treatment option for individuals diagnosed with ACER3 mutation when leukodystrophy is suspected (at the onset) and before severe clinical conditions manifest. In addition, our study opens the way to the structure-based discovery of small molecules able to control the ceramidase activity of ACER3. Ultimately, such modulators could have beneficial

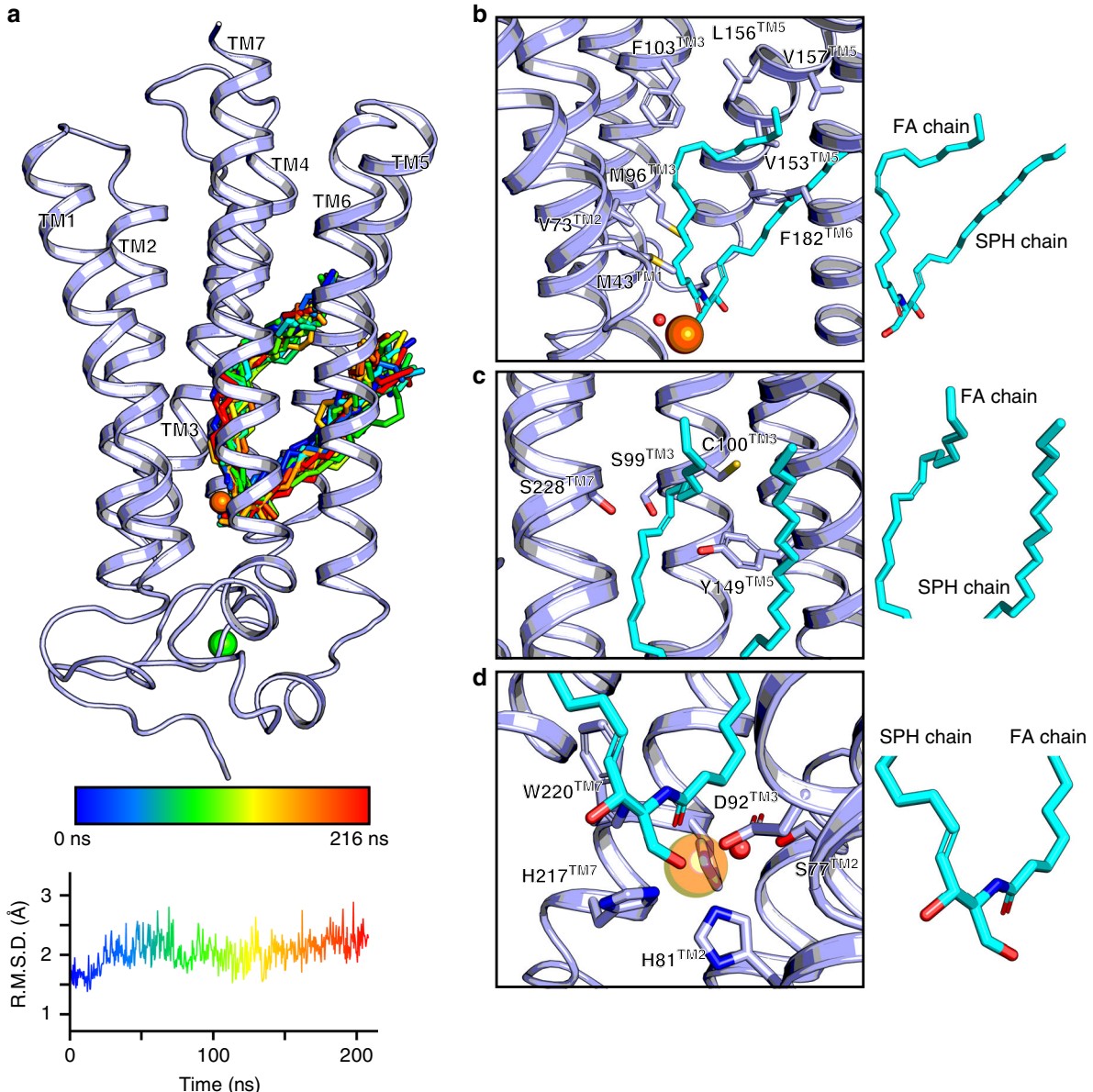

**Fig. 3** Computational analyses of ceramide 18:1 docking into ACER3. **a** Top scoring C18:1 ceramide (shown as stick) docking pose (hydrogens were omitted for clarity) and observed ligand trajectory during a 216 ns long MD simulations (from blue to red with ligand snapshots extracted every 8 ns). A representative all-atom RMSD of the C18:1 ceramide is shown below ($n = 5$ in total). **b** View of the ceramide docking pose highlighting as sticks the residues in close proximity to the ceramide (colored in cyan, and also shown alone on the right to indicate the identity of the fatty acid (FA) and sphingosine (SPH) chains). **c** Environment around the site of unsaturation of the docked C18:1 ceramide with the side chains of polar residues close to the sp$_2$ carbons shown as sticks. **d** Close-up view of the Zn$^{2+}$ catalytic site (side chains represented as sticks) with the putative water molecule (red sphere) ideally positioned to attack the amide bond of the C18:1 ceramide. In all panels, the Zn$^{2+}$ and Ca$^{2+}$ as well as the water molecule are represented as spheres (colored orange, green, and red, respectively)

effects against the pathogenesis of diseases involving C18:1 ceramide and ACER3 dysregulation including colon cancer and AML.

## Methods

**ACER3 contructs.** The full-length synthetic gene of human ACER3 (UniProtKB-Q9NUN7) was subcloned into a modified pFastBac vector resulting in ACER3–BRIL construct bearing a C-terminal BRIL soluble module (thermo-stabilized apocytochrome b562RIL[20]) and an N-terminal Flag epitope followed by a tobacco etch virus protease site. The full-length ACER3–BRIL construct yielded diffraction quality crystals, however, those crystals diffracted to maximum 5 Å resolution despite extensive attempts to optimize the crystallization conditions to improve crystal quality. Therefore, based on an ab initio model of ACER3 obtained from the I-TASSER webserver[30], a putatively flexible C-terminal tail of ACER3 composed of 23 amino acids were truncated which yielded crystals allowing to obtain high-resolution diffraction data. E33G-ACER3 mutant was generated using the full-length ACER3 cloned to pFastBac vector.

**Expression and purification of ACER3 contructs**. ACER3 constructs were expressed in Sf9 insect cells (Life technologies) using the pFastBac baculovirus system (ThermoFisher) according to manufacturer's instructions. Insect cells were grown in suspension in EX-CELL® 420 medium (Sigma) to a density of $4 \times 10^6$ cells per ml and infected with baculovirus encoding ACER3–BRIL. Cells were harvested by centrifugation (3000$g$) 48 h postinfection and stored at −80 °C until purification. After thawing the frozen cell pellet, cells were lysed by osmotic shock in 10 mM Tris-HCl pH 7.5, 1 mM EDTA buffer containing 2 mg ml$^{-1}$ iodoacetamide and protease inhibitors. Lysed cells were centrifuged (38,420$g$) and the enzyme extracted using a glass dounce tissue grinder in a solubilization buffer containing 20 mM HEPES (pH 7.5), 100 mM NaCl, 1% (w/v) n-dodecyl-β-D-maltoside (DDM, Anatrace), 0.1% (w/v) cholesteryl-hemi-succinate (CHS, Sigma), 2 mg ml$^{-1}$

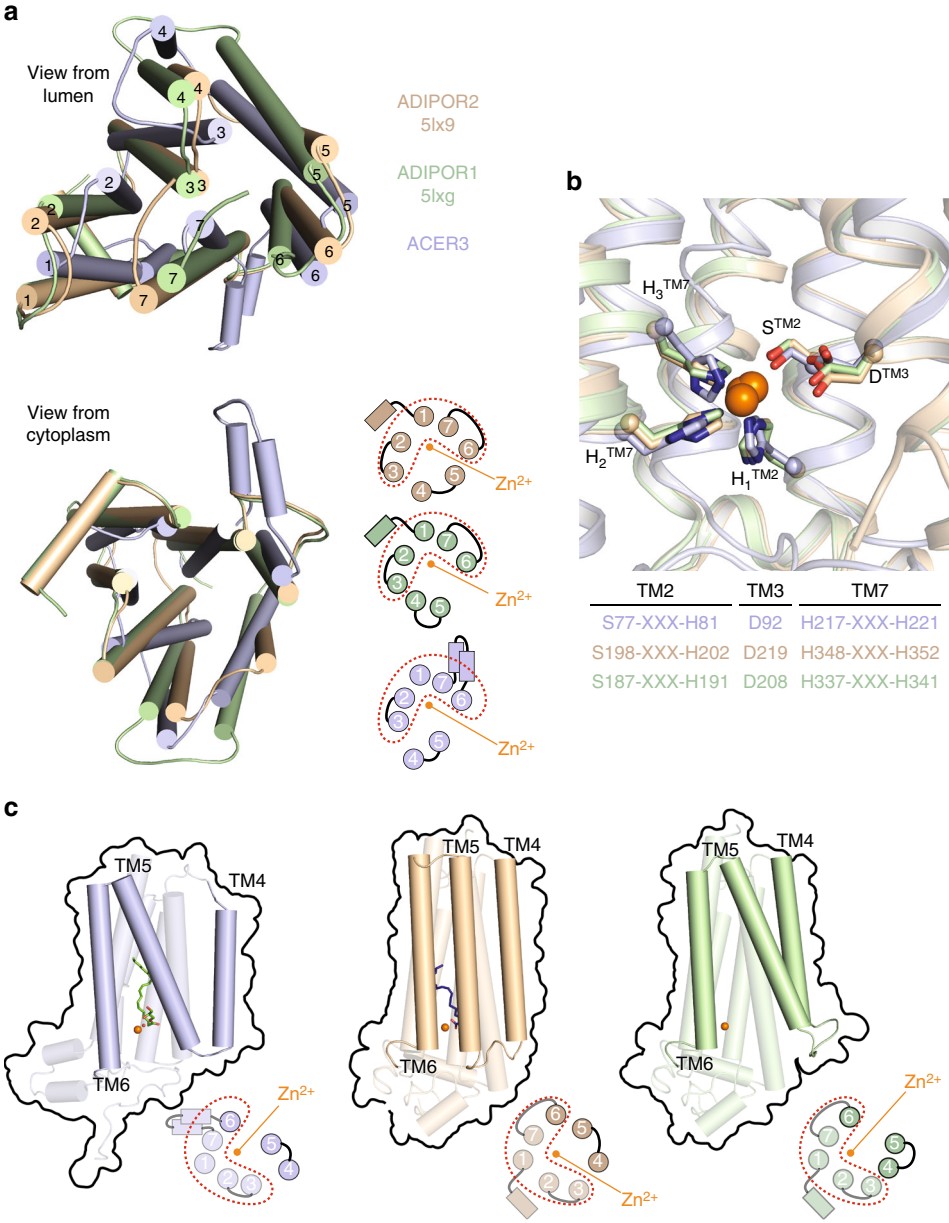

**Fig. 4** Comparison of ACER3 and ADIPORs structures. **a** Superposition of ADIPOR1 (light green), ADIPOR2 (dark yellow), and ACER3 (light blue) represented with cylindrical transmembrane helices viewed from the lumen (top) or from the cytoplasm (bottom). A scheme with numbered TMs is also shown at the bottom to highlight the similar architecture of TM1–3, 6, and 7. **b** Superposition of the three structures showing the strict structural conservation of the $Zn^{2+}$ catalytic site of ACER3, ADIPOR2, and ADIPOR1 (colored as above). The primary sequence with the $H_3SD$ motif is also shown. **c** Views from within the membrane plane showing the differences in the architecture of TM4 and TM5 relative to the $Zn^{2+}$ (orange sphere) between ACER3, ADIPOR2, and ADIPOR1 (colored as described above). The bottom scheme represents views from the cytoplasm, with the red dots highlighting the constant domain. This view also reveals the large TM4 and TM5 shift between the three structures

iodoacetamide and protease inhibitors. The extraction mixture was stirred for 1 h at 4 °C. The cleared supernatant (38,420g centrifugation) was adjusted to the final concentration of 20 mM HEPES (pH 7.4), 200 mM NaCl, 0.5% (w/v) DDM and 0.05% CHS and loaded by gravity flow onto anti-Flag M2 antibody resin (Sigma). The resin was then washed with 2 column volumes (CV) of DDM wash buffer containing 20 mM HEPES (pH 7.5), 200 mM NaCl, 0.025% (w/v) DDM, and 0.0001% (w/v) CHS. While on the M2 antibody resin, the protein was exchanged into lauryl maltose neopentyl glycol (MNG) detergent-containing buffer composed of 20 mM HEPES (pH 7.5), 0.5% MNG-14, 200 mM NaCl. The detergent exchange was performed by washing the column with a series of seven buffers (2 CV each) made up of the following ratios (v/v) of MNG exchange buffer and DDM wash buffer: 1:3, 1:1, 3:1, 9:1, 19:1, 99:1, and MNG exchange buffer alone. The column was then washed with 20× critical micelle concentration (CMC) MNG buffer containing 20 mM HEPES (pH 7.5), 0.02% MNG-14, and 200 mM NaCl and the bound enzyme was eluted in the same buffer supplemented with 0.4 mg ml⁻¹ Flag peptide. The eluted protein was concentrated to 500 µl using 100 kDa spin filters

and further purified by size exclusion chromatography on a Superdex 200 Increase 10/300 column (GE Healthcare) in 20× CMC MNG buffer. Fractions containing monodisperse ACER3-BRIL were collected and concentrated to 20 mg ml⁻¹ for crystallization trials.

**Crystallization, data collection, and processing**. Crystallization of ACER3–BRIL was performed using the in meso method[31]. Concentrated ACER3-BRIL was reconstituted into 10:1 monoolein:cholesterol (Sigma) at a ratio of 1:1.5 protein:lipid by weight. Reconstitution was done using the coupled two-syringe method. The resulting mesophase was dispensed onto a glass plate in 50-nl drops and overlaid with 900 nl precipitant solution using a Gryphon LCP robot (Art Robbins Instruments). Crystals grew in precipitant solution consisting of 34–40% PEG 400, 0.1 M Hepes pH 7.5, 75 mM magnesium sulfate and 5% DMSO. Crystals were observed after one day and grew to full size (~20 µm x 20 µm x 30 µm) after 5 days. Crystals were harvested from the lipidic mesophase using mesh grid loops and directly flash-frozen in liquid nitrogen.

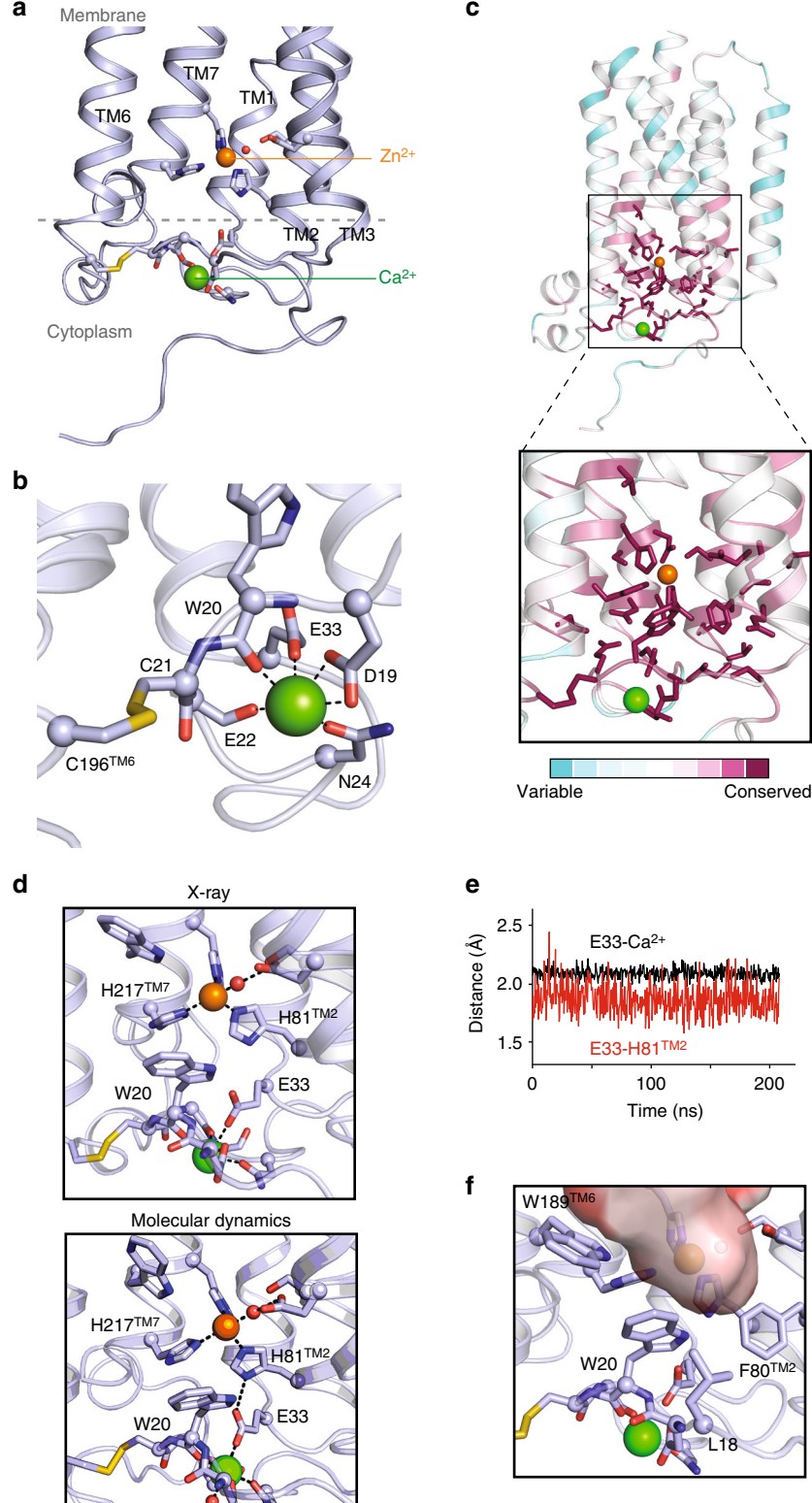

Diffraction data were collected using in the meso crystallographic method[32] at X06SA beamline of the Swiss Light Source (SLS), Villigen, Switzerland. Diffraction measurement from ACER3–BRIL microcrystals were carried out at tolerable dose under cryogenic condition (~100 K) with $10 \times 10$ μm² micro-beam, providing $4.76 \times 10^{11}$ photons/s at 12.4 keV (λ = 1.0 Å, i.e., native) as well as $2.47 \times 10^{11}$ photons/s at 9.67 keV (λ = 1.281 Å, i.e., Zn-edge) X-ray energies. Typically, 10° wedge of data was collected from each microcrystal (from ACER3 BRIL-native and Zn-edge) in oscillation step of 0.2° at a speed of 2°/s using EIGER16M detector with a sample-to-detector distance of 30 cm. A mesh-loop, containing LCP-bolus with many micro-crystals, was raster grid-scanned with microbeam X-ray to locate the crystals. The rastering procedure for crystal detection is described elsewhere[33]. In order to automate identification of tiny crystals, followed by data collection, a new graphical user-interface, called CY + GUI, was developed as an extension of DA + software suite[34] for the beamline at the SLS. Processing of individual small wedges using XDS[35] was automated in DA + software suite. Totally, 198 mini datasets (i.e., 10° wedge/dataset) were collected from ACER3 BRIL native crystals at 12.4 keV X-ray energy. Individual small wedge of dataset measures weak and sparse diffraction spots, which in turn results in partial measurement of each reflection. Thereby, selecting datasets, which are correctly indexed in proper space group, and scaling a subset of statistically equivalent datasets are extremely important to

**Fig. 5** Structure and function of the N-terminus $Ca^{2+}$ binding site of ACER3. **a** Overall view of the N-terminus $Ca^{2+}$ binding site of ACER3 crystal structure from within the membrane plane. Residues participating in $Ca^{2+}$ and $Zn^{2+}$ coordination as well as the disulfide bond formed by C21 and C196$^{TM6}$ are shown as sticks with oxygen, nitrogen, and sulfur atoms red, blue, and yellow, respectively. Crystallographic water molecule next to the $Zn^{2+}$ ion is shown as a red sphere. **b** Arrangement of polar residues and the disulfide bond formed by C21 and C196$^{TM6}$ around the $Ca^{2+}$ ion in ACER3 crystal structure viewed from the intracellular side. The black dashed lines indicate polar contacts. **c** Overall ACER3 crystal structure (left panel) and $Ca^{2+}$ binding site (right panel) are shown as cartoons and colored using the Consurf color scale according to degree of conservation ranging from not conserved, cyan, to highly conserved, magenta. Conserved residues around $Ca^{2+}$ and $Zn^{2+}$ ions (green and orange sphere, respectively) are displayed as sticks. **d** The two residues, W20 and E33, are linking the $Ca^{2+}$ and the $Zn^{2+}$ sites by interacting simultaneously with both metal sites. The top and bottom panels represent the observed structure and the initial pose of the MD simulation, respectively. **e** Panel showing the minimum distances between E33 carboxylate and H81$^{TM2}$ side chain or $Ca^{2+}$ during MD simulations, reflecting the observed hydrogen bond network. **f** Hydrophobic network around W20 side chain forming the bottom part of the putative ceramide pocket

generate a meaningful complete dataset for structural solution. An automated software suite for serial synchrotron crystallographic data selection and merging procedure was developed at the beamline (unpublished, code availability statement: the software will be made available upon request). In order to identify correctly processed datasets, individual mini dataset was first checked for indexing consistency, against user-provided reference unit-cell parameters. Then, 194 datasets, identified as correctly indexed with $a = 60.8\,Å$, $b = 68.8\,Å$, and $c = 257.0$ $Å$ in $C222_1$ space-group, were scaled together using XSCALE[35] program in XDS package. After a first round of scaling, many datasets, which have very low ISa values (described in ref. [36]), were rejected against a cut-off value of 3.0. Thus, 77 datasets were selected out of 194 datasets, and scaled using XSCALE for second round. This yielded a scaled but unmerged dataset with 100% completeness at 2.7 Å resolution. The resolution cutoff was decided based on $CC_{1/2}$ cut-off of 0.3[37]. Similar procedure was applied to 297 mini-wedges of datasets (i.e., 10°/dataset) of ACER3 BRIL–Zn collected at Zn-edge (i.e., $\lambda = 1.281\,Å$). Out of 297 datasets, 198 datasets were selected based on indexing consistency and ISa cutoff of 3.0. These 198 statistically equivalent datasets were scaled together using XSCALE program, yielding a scaled but unmerged dataset with 100% completeness and high multiplicity at 3 Å resolution. Later, these scaled but unmerged HKL files (output from XSCALE) were converted into mtz files for structure determination. Wilson B factors were calculated in XDS.

**Structure determination and refinement**. The structure of ACER3–BRIL was determined by molecular replacement. Initial phases were obtained using the BRIL molecule extracted from PDB 4RWD (chain B) as a search model in PHASER[38]. After density improvement using the CCP4 program parrot[39], the electron density corresponding to ACER3 was still poor and it was only possible to build the C-terminal half of helix 7 (h7$^{1/2}$), which is attached to the BRIL. Subsequently, an ab initio model of ACER3 obtained from the I-TASSER webserver[30] was cut into fragments of two transmembrane helices, which were used as additional search models in PHASER while keeping the BRIL–ACER3 h7$^{1/2}$ fragment as a fixed solution. This procedure allowed us to successfully place two, and then four transmembrane helices. This in turn provided enough phase information to enable manual rebuilding of ACER3 in COOT[40], as well as autobuilding in BUCCANEER[41]. Subsequently, the structure was refined using AUTOBUSTER[42]. At late stages of refinement, translation libration screw-motion parameters generated within AUTOBUSTER were used. MolProbity was used to assess the quality of the structure[43] and indicated that 98% of residues were within favored Ramachandran regions. The data collection and refinement statistics are summarized in Table 1. A stereo image of a portion of the electron density map (including contour level and type of map) is presented in Supplementary Figure 9. Data refinement and analysis software were compiled and supported by the SBGrid Consortium[44]. The Polder OMIT map[23] and the zinc difference anomalous map where calculated in PHENIX[45].

**Docking calculations**. Computational docking of N-Oleoyl-D-sphingosine (d18:1/ 18:1) (C18:1 ceramide) was performed with the program Protein-Ligand ANT System (version 1.2) using as a receptor our 2.7 Å structure in which all nonprotein atoms except zinc and calcium were removed. PLANTS combines an ant colony optimization algorithm with an empirical scoring function for the prediction and scoring of binding poses in a protein structure[46]. Ten poses were calculated and scored by the plp scoring function at a speed setting of one. Each of the ten ligand poses actually represents the best scoring pose from a cluster of solutions. The number of iterations necessary for adequate sampling of the ligand (and optionally receptor) degrees of freedom is determined automatically by the program and for this particular system was 1387, resulting in about $5 \times 10^7$ scoring function evaluations. The binding pocket of ACER3 was defined by all residues within a 25 Å radius around the zinc atom. All other options of PLANTS were left at their default settings. The top scoring pose was selected and used as input for MD simulations.

Of note, similar top scoring C18:1 ceramide binding poses were obtained using GlideXP as well as the SWISSDOCK webserver (in blind docking mode) (Supplementary Figure 5c).

**System preparation and MD simulations**. Three protein-ligand systems were subjected to MD simulations. We initially performed two independent simulations of unliganded ACER3 (i.e., the crystal structure in which the BRIL moiety and the monoolein were deleted—system 1) of about ~300 ns. However, in this system the absence of a bound lipid resulted in rapid water influx in the zinc and lipid binding cavity and led to destabilization of the zinc binding site (and zinc unbinding). This system was not analyzed further. Subsequently, we used a model of ACER3 in complex with the top scoring ceramide–C18:1 (d18:1/18:1) docking pose (system 2). The crystallographic zinc and calcium ions, as well as three ordered water molecules located close to the zinc binding site were included in the system. The third system was identical to system 2 except that E33 was mutated to a glycine. The WT ACER3–ceramide–C18:1 and E33G mutant MD systems were then set up using the CHARMM-GUI membrane builder[47]. The proteins were inserted into a hydrated, equilibrated bilayer composed of 80 molecules of 2-oleoyl-1-palmitoyl-sn-glycero-3-phosphocholine (POPC) and 20 molecules of cholesterol in the upper leaflet, and 115 molecules of ceramide–C18:1 in the lower leaflet in order to simulate in the presence of high concentration of substrate. Similar observations were obtained in MD trajectories performed in membranes composed of roughly 116 POPC, 44 POPE, 6 sphingomyelin, 14 POPI, and 14 POPS molecules (equally partitioned between the two leaflets). This composition should be close to the endoplasmic reticulum membrane[48]. These MD trajectories are available upon request.

Totally, 32 sodium and 38 chloride ions were added to neutralize the system, reaching a final concentration of approximately 150 mM. Topologies and parameters for ceramide–C18:1 (d18:1/18:1) were available in the additive all-atom CHARMM lipid force field[49].

MD calculations were performed in in GROMACS 2016.3 using the CHARMM36 force field and the CHARMM TIP3P water model. The input systems were subjected to energy minimization, equilibration, and production simulation using the GROMACS input scripts generated by CHARMM-GUI[50]. Briefly, the system was energy minimized using 5000 steps of steepest descent, followed by 375 ps of equilibration. NVT (constant particle number, volume, and temperature) and NPT (constant particle number, pressure, and temperature) equilibrations were followed by NPT production runs for both systems. The van der Waals interactions were smoothly switched off at 10–12 Å by a force-switching function[51], whereas the long-range electrostatic interactions were calculated using the particle mesh Ewald method[52]. The temperature and pressure were held at 310.15 K and 1 bar, respectively. The assembled systems were equilibrated by the well-established protocol in Membrane Builder, in which various restraints were applied to the protein, lipids, and water molecules, and the restraint forces were gradually reduced during this process. During production simulations an NPT ensemble was used with semi-isotropic pressure coupling via the Parrinello–Rahman barostat method while the Nose–Hoover thermostat was used to maintain a temperature of 310.15 K (as described in ref. [21]). A leapfrog integration scheme was used, and all bonds were constrained allowing for a time-step of 2 ps to be used during NPT equilibration and production MD simulations. For both wild type and E33G mutant systems, we performed five independent production runs of ~ 200 ns each. Production runs were subsequently analyzed using GROMACS tools to yield RMSD and root mean square fluctuations.

**Ceramidase activity assay**. The enzymatic activity of ACER3–BRIL was monitored by sphingosine detection and quantification using liquid chromatography (LC) MS/MS analyses. All solvents and reagents used were of highest available purity and purchased from typical suppliers. The D-erythro-sphingosine (d18:1), ceramide C18 (d18:1/18), and ceramide C18:1 (d18:1/18:1) used in this study were synthesized according to previous methods developed by the Arenz laboratory[53]. All other lipids were purchased from Avanti Polar Lipids. Mutants were generated by site-directed mutagenesis, confirmed by DNA sequencing (MWG-Eurofins) and expressed and purified as described for the ACER3–BRIL construct. The list of primers used in this study are listed in the Supplementary Table 1.

Ceramidase activity assays were performed by incubating purified ACER3–BRIL WT or mutants (1 μM) with ceramide (20 μM) for 1 h at room temperature in 20 mM HEPES (pH 7.5), 200 mM NaCl, 0.025% (w/v) DDM, and

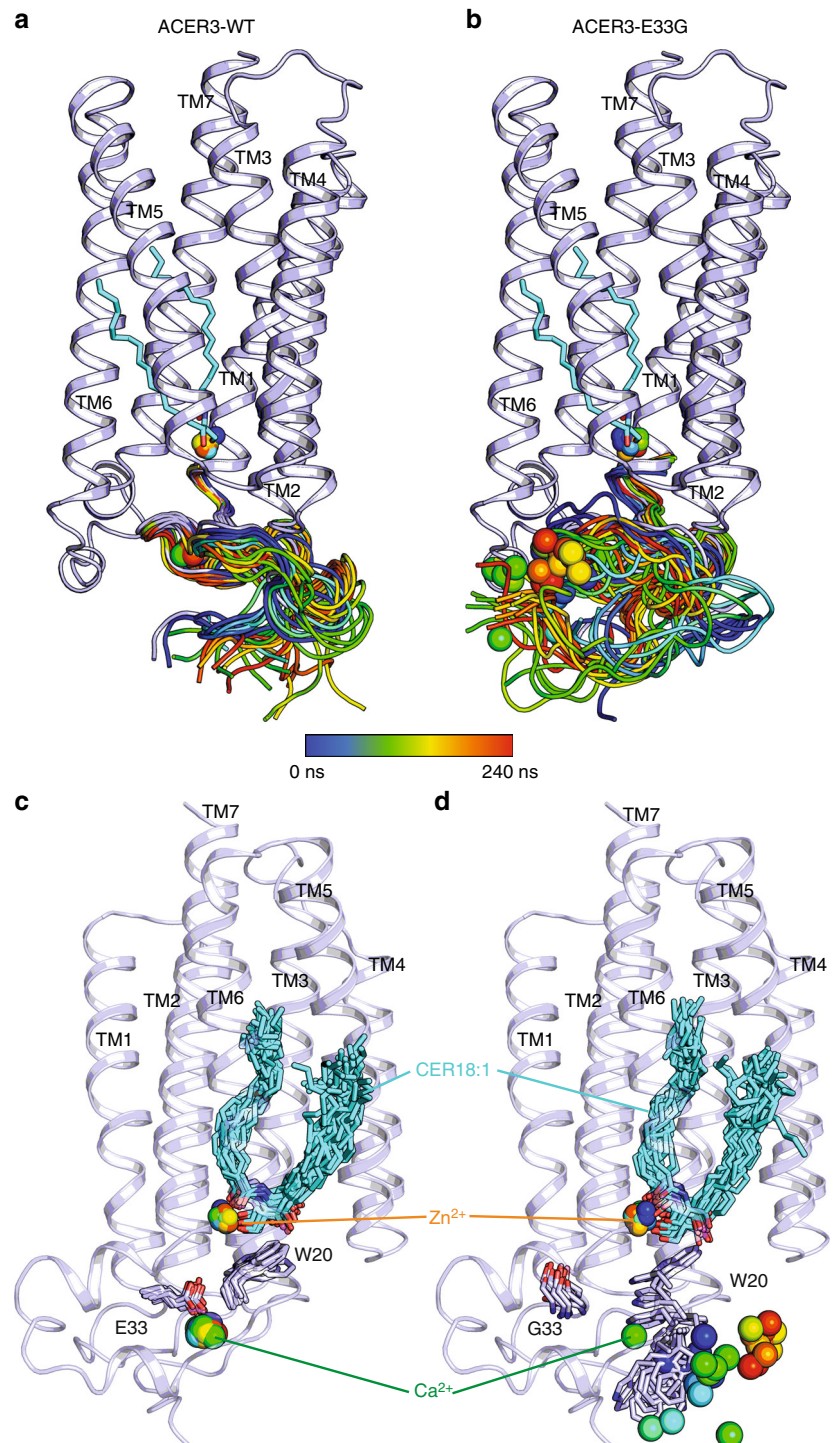

**Fig. 6** MD simulations of ACER3 wild-type and E33G mutant Snapshots of the calculated N-terminus domain trajectories during a 240 ns long MD simulation (from blue to red, taken every 10 ns) for the wild-type (**a**) or E33G mutant (**b**) showing that the mutation leads to a more dynamic/unstable N-terminus domain. **c**, **d** Snapshots of the ACER3 wt (**c**) or E33G mutant (**d**) showing the key ligands (Cer18:1, $Zn^{2+}$ and $Ca^{2+}$) and contacting residues (shown as sticks) trajectories during a 200 ns long MD simulations colored as above and highlighting the motion/instability of the W20 side chain and of the $Ca^{2+}$ in the E33G mutant (**d**) in contrast to the wild-type ACER3 (**c**)

0.0001% (w/v) CHS. Reactions were quenched by addition of methanol (final concentration 30%) which contained 68.5 pg of the internal standard, sphingosine (d17:1).

Lipids were extracted from reaction samples using the previously reported method[54]. Briefly, a mixture of dichloromethane/methanol (2% acetic acid)/water (2.5:2.5:2 v/v/v) was added to each reaction and the solution was centrifuged. The organic phase was collected and dried under nitrogen, then dissolved in 10 μL of MeOH. The lipid extract was stored at −20 °C before LC–MS/MS analysis.

LC-MS/MS analysis of formed sphingosine was performed using an Agilent 1290 Infinity. The samples were separated on an Acquity UPLC BEH-C8 column (particle size 1.7 μm, 2.1 × 100 mm) (Waters) maintained at 35 °C. The mobile phases consisted of eluent A (99.9% $H_2O$: 0.1% HCOOH, v/v) and eluent B (99.9% $CH_3CN$: 0.1% HCOOH, v/v). The gradient was as follows: 50% B at 0 min, 60% B at 2 min, 60% B at 3 min, 100% B at 4 min, 100% B at 8.5 min and 50% B at 9 min. The flow rate was 0.3 mL/min. The auto sampler was set at 5 °C and the injection volume was 5 μL. The HPLC system was coupled on-line to an Agilent 6460 triple

quadrupole MS equipped with electrospray ionization source operated in positive ion mode. The source parameters used were as follows: source temperature was set at 300 °C, nebulizer gas (nitrogen) flow rate was 10 L/min, sheath gas temperature was 300 °C, sheath gas (nitrogen) flow rate was 12 L/min and the spray voltage was adjusted to +4000 V. The collision energy optimums for sphingosine (d17:1) and sphingosine (d18:1) were 5 eV. Analyses were performed in selected reaction monitoring detection mode using nitrogen as collision gas. Peak detection, integration and quantitative analysis were done using MassHunter QqQ Quantitative analysis software (Agilent Technologies) and Microsoft Excel software.

Sphingosine (d18:1): precursor ion $m/z = 300.5$ and product ion $m/z = 282.5$.
Sphingosine (d17:1): precursor ion $m/z = 286.5$ and product ion $m/z = 268.5$.

The presented data for enzymatic activity are representative of three experiments performed on three independent ACER3–BRIL enzyme preparations and two independent mutants' preparations, each experiment contained five or six replicates.

**TR-FRET experiments in living cells**. YFP-ADIPOR2 was kindly provided by the laboratory of Dr. Vazquez-Martinez. For the ADIPOR2-YFP construct, full-length AdipoR2 was inserted in frame of the N terminus of the YFP in pcDNA 3.1-YFP by PCR using the following specific oligonucleotides: forward 5′ TAAGCAGCTAGC ATGAACGAGCCAACAGAAAACC 3′ and reverse 5′ TAAGCAAAGCTTCA GTGCATCCTCTTCAC 3′. Full-length ACER3 was inserted in frame of the N terminus of the SNAP tag in the pSNAPf vector (New England Biolabs) by PCR using the following specific oligonucleotides: 5′ AAACGCTAGCGATATCATGG ACTACAAGGACGACGACGACAAGGCTCCTGCAGCTGACAG 3′ as forward and 5′ TATGCAACCGGTGTGCTTCCTCAAGGGCT 3′ as reverse.

HEK293 cells were grown in Dulbecco's Modified Eagle Medium supplemented with 10% fetal bovine serum and antibiotics at 37 °C, 5% $CO_2$. Lipofectamine 2000-based reverse transfection were performed in polyornithine coated black-walled, dark-bottom, 96-well plates. We used 30 ng of ACER3–SNAP tag with either 50 ng of YFP-ADIPOR2, ADIPOR2-YFP, or empty PRK6 as control and $10^5$ cells per well.

After 24 h transfection cells were fixed with 3.6% PFA and permeabilised with 0.05% Triton X100. Then, intracellular ACER3–SNAP was labeled with BG-Lumi4-Tb (300 nM) for 1 h at 37 °C and after several washes, YFP and lumi4 fluorescence signals were observed on an Infinite 500 (Tecan, Seestrasse, Switzerland). The signal was collected both at 520 nm and 620 nm (10 flashes, 400 μs integration time, 150 μs lag time). Background values from nontransfected cells were removed from raw data in each channel. Subsequently, the normalized time resolved FRET ratio was obtained by dividing the specific acceptor signal at 520 nm by the specific donor signal at 620 nm data and plotted using GraphPad Prism (GraphPad Software, Inc., San Diego, CA).

**Code availability**. An automated software suite for serial synchrotron crystallographic data selection and merging procedure was developed at the beamline (unpublished, the software will be made available upon request).

**Reporting summary**. Further information on experimental design is available in the Nature Research Reporting Summary linked to this article.

## Data availability
Data supporting the findings of this manuscript are available from the corresponding authors upon reasonable request. Coordinates and structure factors for the ACER3-BRIL structure have been deposited in the Protein Data Bank under accession number 6G7O.

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

## Acknowledgments

We are grateful to Vincent Oliéric and Takashi Tomikazi from PSI for providing assistance in using beamline X06SA. We thank the MetaToul-Lipidomique Core Facility (I2MC, Inserm 1048, Toulouse, France), MetaboHUB-ANR-11-INBS-0010 for the sphingosine quantification experiments. The research leading to these results has received funding from the European Union's Horizon 2020 research and innovation program under grant agreement n.° 730872, project CALIPSOplus. This project has received funding from the European Research Council (ERC) under the European Union's Horizon 2020 research and innovation program (grant agreement N° 647687).

## Author contributions

IVB expressed, purified, characterized and crystallized ACER3-BRIL preparations with the help of P.R. and T.W.A. R.D.H. expressed and purified ACER3-BRIL and mutants, optimized and performed all the enzymatic assays with the help of T.W.A., J.S.P., and I.V.B. M.F. performed the TR-FRET experiments. E.M.S. synthesized ceramides of different chain lengths and performed HPLC-MS analysis of the ceramide cleavage reactions. C.A. supervised E.M.S. R.D.H., C.L., C.G., collected data with the help of F.H. C.L. performed the computational studies with help from C.G. S.B. helped with the X-ray data acquisition and processing. C.L. solved and refined the structure. R.S. prepared the figures. S.G. wrote the manuscript with the help of R.S., C.L., I.V.B., and R.D.H. All authors contributed to the manuscript preparation. S.G. designed research and supervised the overall project.

## Additional information

**Competing interests:** The authors declare no competing interests.

