## [Peer Review File · Nature Communications]

Reviewers' Comments:

Reviewer #1:

Remarks to the Author:

The article reports the crystal structure of the intramembrane protein alkaline ceramidase 3 (ACER3). ACERs breakdown ceramides to regulate several metabolic pathways and pathologies including cancer, and ACER3 is notable for its mutation in leukodystrophy. Previous crystal structures of ceramidases from the acidic and neutral subtypes have been determined and the structure of the ACER3 basic subtype reported here, completes the set. The structure reveals a 7TM protein that is structurally similar to adiponectin receptors and appears to use Zn and Ca ions to catalyze substrate hydrolysis. The main novel finding is the calcium-binding site that likely regulates the enzyme's function. Overall, the paper is well written and the results are clearly described and nicely illustrated. However, there are several grammatical errors scattered throughout the paper (some of which are indicated below) that should be corrected.

The main weakness of the paper is the lack of biochemical/in vivo validation of hypotheses derived from the structure. An in vitro activity assay showing enzyme activity was included (Extended Fig. 1b), so I am surprised the authors did not at the very least, test some mutants in this in vitro assay. E33G provides a nice naturally occurring mutation, but other mutants would strengthen the paper: for example, hydrophobic residue mutations in the ceramide binding tunnel to validate ceramide docking; S99, Y149, S228 to distinguish C18:1 vs C18:0 binding, and most importantly other Ca binding residues in addition to E33.

Additionally, how can the authors be sure that it is indeed a Ca ion bound to the enzyme in their crystal given that there is 75 mM Mg-sulfate in the crystallization condition?

Minor issues:

Line 70:

Change cystein to cysteine.

Line 106:

'impossible' is too strong, maybe change to 'more difficult'

Line 112:

"results in the calcium binding site destabilization" TO "results in destabilization of the calcium binding site"

Lines 121-125:

Do the truncated 23 C-terminal residues have any predicted function? That segment contains 6 cationic residues and 15 hydrophobic ones; could it be interacting with the membrane? Has the enzymatic activity of the C-terminally-truncated protein fused to BRIL been compared with the activity of full-length wild-type?

Line 138:

Remove "The" from "The ICL3"

Line 145:

How is a disulfide bridge in the reducing environment of the cytoplasm maintained? Is it buried?

Line 157:

The hydrophilic patch and water molecule seem to be located near the ends of the docked ceramide acyl chains, opposite from the ceramide polar groups?

Line 236:

"common with the ACER3 structure..."

Line 246:

"the N-terminal domain"

Lines 309-314:

As the calcium ion is coordinated by carbonyls and by aspartates/glutamates with pKa values presumably around 4, and the site is facing the cytoplasm, is pH expected to have an influence on calcium binding?

Line 369:

Were 100kDa or 10kDa spin filters used?

Line 471:

What was the reason for setting up an all-ceramide membrane leaflet?

Figure 1a:

It is not clear to which TM some of the TM# labels are attached.

Figure 3d:

The relative positions of ceramide and the zinc ions are not clear. The view should be rotated.

Extended data figure 3e:

The distances to the zinc ion should be indicated.

Extended data figure 4a:

The zinc- and calcium-binding residues should be marked on/above the alignment.

A citation for the ConSurf server should be added in the references.

Extended data figure 4b is not really useful and could optionally be removed.

Extended data figure 5a:

The distances between ceramide and W220/Zn/water molecule should be indicated in panel 1.

Extended data figure 5a legend:

Does the zinc ion also serve as oxyanion hole for the amide carbonyl oxygen, along with W220?

Extended data figure 7b and c:

The distances to the calcium ion should be added, maybe in the bottom schematics.

Extended data table 1:

- The redundancy-independent merging R factor (Rmeas) should be provided instead of Rmerge, due to the high redundancy of the datasets.
- The Wilson B-factor should be added to the data collection section
- The R-work and R-free are unusually close in value; an explanation should be provided for this. Could it be that the refinement was carried out for an insufficient number of cycles after the last manual changes to the model?
- In the structure validation report, the clash score is high (23); typical values are below 5 for a medium resolution structure. The clashes listed in the table on page 9 should be inspected. This is also often improved by adding hydrogen atoms to the model and refining for many cycles.
- The number of side chain outliers (6.5 %) is also high, with typical values around 1 %. The rotamer outliers in the table on page 13 should be inspected.

Reviewer #2:

Remarks to the Author:

The manuscript by Vasiliauskaitė-Brooks et al describes the first crystal structure of human alkaline ceramidase 3 (ACER3) to reveal a 7-TM architecture and active site similar to the adiponectin receptors. The structure provides insight into catalysis and a point mutation involved in disease. The manuscript is important and clearly written. There are some points that need to be clarified.

Major Points

1. There is concern with the accuracy of the modeled monoolein and whether it should be included in the refined structure given the ambiguity of the ligand identity. From the provided figure, it appears the polar section of monoolein resides in a highly hydrophobic environment, the electron density appears to extend beyond the polar headgroup section (ext Fig 3d), and there is a break in the electron density between the polar and hydrophobic sections near the Zn ion. Furthermore, the biological insight from modeling this is minimal.
2. Substrate specificity. The proposed steric hindrance mechanism for ceramide specificity versus sphingomyelin and glycosylceramide appears reasonable. However, the figure and text does not shown clearly what residues are blocking binding of larger lipids or how far the pocket extends below the ceramide headgroup.
3. The calcium and magnesium ions. Given Magnesium is present in the crystallization conditions, there should be some evidence or rationale included as to why the calcium ion is modeled as a calcium ion versus a magnesium ion. Secondly, where the magnesium ions bind in the crystal lattice is not shown, even though two are included in the refinement statistics.

Minor points

4. In extended data Fig. 2c, it is not clear what YFP-AdipoR2 and AdipoR2-YFP were co-expressed with. The assumption is ACER3-SNAP, but additional detail in the figure or figure legend would help clarify.
5. In Fig. 1B and 1C, please label the Ca ion for clarity. Also, the orientation to the membrane is lost in Fig. 1B, which would help identify the position of the Arg and Lys residues for putative anionic lipid binding.
6. In fig. 2, can the electron density of the placed water molecule be shown?
7. In Fig. 2, where is the intramembrane pocket accessible to the lipid leaflet and can you label these sites? In figure 3, it looks like the lower pocket (that accommodates the sphingosine moiety) is accessible, but is the upper pocket (where the acyl chain is proposed to bind) also accessible the membrane?
8. Figure 3. Label sphingosine and acyl chains of ceramide for clarity.
9. How will the modeled ceramide poses in ACER3 be made available to the scientific community? A pymol session?
10. Should the title specify that this is alkaline ceramidase 3?

Reviewer #3:

Remarks to the Author:

In this study, an X-ray crystal structure of human alkaline ceramidase type 3 (Acer3) at 2.7 Å resolution is reported. The enzyme is shown to consist of 7 transmembrane helices and to be a zinc hydrolase. Its regulation by calcium is explained on the basis of the crystal structure and a molecular rationale for the loss of function associated with mutants in progressive leukodystrophy is provided. The findings will be of interest to the membrane structural and functional biology communities as well as those interested in structure-based drug discovery and lipid signalling and metabolism. The work is done competently and the conclusions are reasonable and soundly based for the most part.

The following points should be addressed in a revision of the manuscript.

Describe the clinical phenotype of leukodystrophy and the ACER 3 E33G mutant in humans. Does a treatment exist for the disease?

"The molecular control of ACER enzymatic activity (agonists and antagonists) thus appears as a possible clinical intervention for the treatment of..."

This is an unsatisfactory statement. Advocating the possible use of antagonists and agonists in different diseases suggests that both types will have side effects that limit their utility. Accordingly, more research will be needed to show that modulation of ACER3 is a reasonable clinical intervention. The authors should rewrite this section placing less emphasis on clinical relevance and address the difficulties in developing drugs due to the broader implications associated with modulating ceramide homeostasis.

Explain how the lipid used in the crystallization trials amount to 54% by weight. The concentration of monoolein is at least a thousand times higher than reported. Show the revised calculations.

Was a co-crystal structure of ACER3 with any sphingosine lipid attempted? This is an obvious thing to do and should be addressed in the manuscript.

The orientation of the monoolein in the putative binding pocket would appear to differ from the proposed substrate-binding pose. The authors should comment on this.

ED Fig. 5a. It is hard to distinguish covalent bonds from polar interactions. The difference should be obvious in a revised version of the figure.

Did the authors perform MDS using ceramide containing the less preferred 18:0 acyl chain? It seems logical to perform this simulation to address the substrate specificity issues raised in the manuscript.

State how the identity of calcium was verified in the crystal structure and provide citations to the effect calcium has on ceramidase activity.

The authors describe and compare the different conformations of ADIPORs and ACERs, with a focus on TM4 and TM5. For the edification of the general reader, the process mediated by the conformational change should be explained. The authors refer to open and closed states. Does this mean the conformational change enables substrate binding and/or product release at ADIPORs? Would ACER3 be able to undergo a conformational change similar to that undergone by ADIPORs? The authors should address the issue of substrate access to the putative binding pocket in ACERs.

The loss of ceramidase activity in E33G ACER3 mutants might well originate earlier in the enzyme's path to maturity. Specifically, the E33G mutation is likely to eliminate calcium binding and early folding. The protein may never make it to its destination membrane. This issue should be addressed.

Include a single pose of ceramide somewhere in Fig. 3a where its parts can be clearly seen and

understood by the reader.

We would like to thank the referees for the constructive comments. The manuscript has been revised to address their main concerns and comments. Below, we provide a point-by-point response. Each referee's comments are in coloured fonts and our response is in normal black font. Text changes made in the manuscript are in italic underlined fonts.

Reviewer #1 (Remarks to the Author):

The article reports the crystal structure of the intramembrane protein alkaline ceramidase 3 (ACER3). ACERs breakdown ceramides to regulate several metabolic pathways and pathologies including cancer, and ACER3 is notable for its mutation in leukodystrophy. Previous crystal structures of ceramidases from the acidic and neutral subtypes have been determined and the structure of the ACER3 basic subtype reported here, completes the set. The structure reveals a 7TM protein that is structurally similar to adiponectin receptors and appears to use Zn and Ca ions to catalyze substrate hydrolysis. The main novel finding is the calcium-binding site that likely regulates the enzyme's function. Overall, the paper is well written and the results are clearly described and nicely illustrated. However, there are several grammatical errors scattered throughout the paper (some of which are indicated below) that should be corrected.

We are sorry for the grammatical errors, and these have now been corrected throughout the manuscript as requested by Reviewer #1.

The main weakness of the paper is the lack of biochemical/*in vivo* validation of hypotheses derived from the structure. An *in vitro* activity assay showing enzyme activity was included (Extended Fig. 1b), so I am surprised the authors did not at the very least, test some mutants in this *in vitro* assay. E33G provides a nice naturally occurring mutation, but other mutants would strengthen the paper: for example, hydrophobic residue mutations in the ceramide binding tunnel to validate ceramide docking; S99, Y149, S228 to distinguish C18:1 vs C18:0 binding, and most importantly other Ca binding residues in addition to E33.

The reviewer #1 is right that a mutagenesis study has to be performed to validate some of our hypothesis. The main hypothesis we derived from the structure is about the role of E33G in ACER3 function. As already mentioned in the main text, functional studies on purified material were rendered technically challenging by the instability of this mutant in detergent. However, a recent published study by Edvardson *et.al.* clearly demonstrated *in vivo* and *in vitro* (using both patients' cells and microsomes from yeast strains devoid of an intrinsic ceramidase activity) that the mutant E33G leads to a complete loss of ceramidase activity thus providing solid functional data to support our main conclusion.

The main purpose of including the activity data we present is to validate the crystallography construct remains functional (and substrate specificity is maintained) after purification in detergents, and thus the structure gleaned is biologically relevant.

The structure-function analysis of ACER3 is undergoing, and we appreciate the reviewer's comments highlighting the importance of such biochemical data to analyze ceramide docking, ceramide specificity and other important enzymatic parameters. However we strongly believe that these studies do not fall within the breadth of this study and will justify

the preparation of a separate manuscript. Indeed, a mutagenesis approach is not necessarily straightforward, and we expect that several mutants, in particular those for Ca²⁺ binding, would be unstable during purification. We could use yeast microsomes fraction as described above. However, there is a significant drawback using this system as yeast microsomes preparations are significantly distinct in term of lipidic environment when compared to membranes derived from human cells. We are thus trying to develop a functional assay using plasma membranes from human origins (such as HEK cells) transfected with ACER3 constructs. This work will take some time before the assay is validated and can be used for a thorough functional analysis.

Additionally, how can the authors be sure that it is indeed a Ca ion bound to the enzyme in their crystal given that there is 75 mM Mg-sulfate in the crystallization condition?

This is an excellent question that was raised by the three reviewers. In fact, we did perform structure refinement with both Mg²⁺ and Ca²⁺ over the course of solving the ACER3 structure. In the presence of Mg²⁺ the calculated Fo-Fc map clearly indicated that some electrons were missing (positive peak in the Fo-Fc map). We apologize for not having included this information in the first version of the manuscript. To make this important point clear for the reader we are now showing the results of the Fo-Fc maps calculated either with Ca²⁺ or with Mg²⁺ in Extended data Figure 7. These difference maps, together with the average oxygen-metal distance of ~ 2.3 Å and the observed coordination geometry, unambiguously indicate that the metal present in this site is indeed a calcium ion.

Minor issues:

Line 70:

Change cystein to cysteine.

Done.

Line 106:

'impossible' is too strong, maybe change to 'more difficult'

Done.

Line 112:

"results in the calcium binding site destabilization" TO "results in destabilization of the calcium binding site"

Done.

Lines 121-125:

Do the truncated 23 C-terminal residues have any predicted function? That segment contains 6 cationic residues and 15 hydrophobic ones; could it be interacting with the membrane? Has the enzymatic activity of the C-terminally-truncated protein fused to BRIL been compared

with the activity of full-length wild-type?

Those are interesting questions and, as described above, they will be part of a further thorough functional study of ACER3 using a more biologically relevant system (HEK cells). Wild-type ACER3 activity is known to show a substrate specificity to C18:1 over C18:0, our enzymatic data confirms that our construct retains this specificity. The main point, for our structural study, was to demonstrate that the BRIL module does not impair the enzymatic function such that the observed structure is, at least in part, functionally relevant.

Line 138:

Remove "The" from "The ICL3"

Done.

Line 145:

How is a disulfide bridge in the reducing environment of the cytoplasm maintained? Is it buried?

This is an interesting question. This disulfide does not seem to be buried. The literature suggests however that cysteine residues may exist as "thiolate anions at neutral pH due to a lowering of their pKa values by charge interactions with neighboring amino acid residues and are therefore more vulnerable to oxidation" (Cumming et.al. The Journal of Biological Chemistry 279, 21749-21758). A sentence was added to alert the reader on this particular point:

The crystal structure uncovers a disulfide bond formed between C21 and C196 connecting the N-terminus to TM6. This disulfide may play an important role in stabilizing this peculiar N-terminal domain fold (Fig. 1c). In a cellular context, this disulfide is facing the reducing environment of the cytoplasm and we cannot exclude the possibility that it is dynamically regulated by its redox local environment, in particular neighboring charged residues/ lipids may lower thiolate pKa and render those more prone to oxidation (22).

Line 157:

The hydrophilic patch and water molecule seem to be located near the ends of the docked ceramide acyl chains, opposite from the ceramide polar groups?

The referee is right. In fact, we meant that the hydrophilic patch might play a role in the selectivity of ceramide binding from the acyl chain length point of view. Indeed, this patch might limit the binding of long acyl chain ceramides (over 24). Again, at this stage, this is a

speculative hypothesis and functional data would be required to validate this point.

We added a new sentence to clarify this point in the main text:

Such a hydrophilic patch might play a role in ceramide binding selectivity, as it may prevent the binding of ceramide harboring acyl chains over 24 carbons.

Line 236:

“common with the ACER3 structure...”

Done.

Line 246:

“the N-terminal domain”

Done.

Lines 309-314:

As the calcium ion is coordinated by carbonyls and by aspartates/glutamates with pKa values presumably around 4, and the site is facing the cytoplasm, is pH expected to have an influence on calcium binding?

The pH will definitively not directly influence the calcium binding to carbonyls and side chains of coordinating residues. However, we do believe that it could happen indirectly through a change in local Ca^{2+} concentration. Indeed, it was previously shown that pH can significantly modify the membrane calcium binding in particular through phospholipids (Langer, G.A, Circ Res. 1985;57:374-382). It was proposed at that time that the pH might influence the charge of NH_2 groups that are shielding negatively charged Ca^{2+} binding sites. Thus, the observed pH dependence *in vitro*, could originate from such effects. We added the reference to this study in the text.

Line 369:

Were 100kDa or 10kDa spin filters used?

We are using 100 kDa spin filters, as the detergent micelles significantly contribute to the size of our preparations. Together with the BRIL, it corresponds to objects that are well over 100 kDa in mass.

Line 471:

What was the reason for setting up an all-ceramide membrane leaflet?

The rationale for setting up the simulation using an all-ceramide lower leaflet was to simulate the protein in the presence of a high concentration of substrate. We have added this point in the method section:

20 molecules of cholesterol in the upper leaflet, and 115 molecules of ceramide-C18:1 in the lower leaflet *in order to simulate in the presence of high concentration of substrate.*

Moreover, we have produced additional M.D. trajectories in membranes composed of roughly 53% POPC, 20% POPE, 3.7% sphingomyelin, 7% POPI and 7% POPS. This composition should be close to the ER membrane based on information available at the membrane protein lipid composition atlas (<http://opm.phar.umich.edu/atlas.php?membrane=Endoplasmic%20reticulum%20membrane>). However, we did not observe any major differences in the conformation and dynamics of ACER3, the ceramide ligand or the metal ions. This observation and the availability of the new simulations have been indicated in the method section of the revised manuscript.

Similar observations were obtained in MD trajectories performed in membranes composed of roughly 53% POPC, 20% POPE, 3.7% sphingomyelin, 7% POPI and 7% POPS. This composition should be close to the ER membrane based on information available at the membrane protein lipid composition atlas (<http://opm.phar.umich.edu>)

These MD trajectories are available upon request.

In the near future, it will be interesting to use enhanced sampling techniques to simulate the binding/unbinding of ceramide using both setups. However, such binding/unbinding events are beyond the time scale of the current classical M.D. simulations.

Figure 1a:

It is not clear to which TM some of the TM# labels are attached.

The figure has been modified to make this clearer.

Figure 3d:

The relative positions of ceramide and the zinc ions are not clear. The view should be rotated.

The figure has been modified to make this clearer.

Extended data figure 3e:

The distances to the zinc ion should be indicated.

The figure has been modified to make this clearer.

Extended data figure 4a:

The zinc- and calcium-binding residues should be marked on/above the alignment.

The figure has been modified to make this clearer.

A citation for the ConSurf server should be added in the references.

Done.

Extended data figure 4b is not really useful and could optionally be removed.

We felt that it could be used by some readers and decided to leave it.

Extended data figure 5a:

The distances between ceramide and W220/Zn/water molecule should be indicated in panel 1.

The figure has been modified to indicate the distances.

Extended data figure 5a legend:

Does the zinc ion also serve as oxyanion hole for the amide carbonyl oxygen, along with W220?

Yes, it does. We have added a sentence in the legend to clarify this point:

W220 side chain polarizes the amide carbonyl and, together with the Zn^{2+} , stabilize the oxyanion formed in the tetrahedral transition state (2).

Extended data figure 7b and c:

The distances to the calcium ion should be added, maybe in the bottom schematics.

The figure has been modified to indicate the distances.

Extended data table 1:

- The redundancy-independent merging R factor (Rmeas) should be provided instead of Rmerge, due to the high redundancy of the datasets.

Rmeas has been added to EDT1.

- The Wilson B-factor should be added to the data collection section

It has been added as requested.

- The R-work and R-free are unusually close in value; an explanation should be provided for this. Could it be that the refinement was carried out for an insufficient number of cycles after the last manual changes to the model?

The R-work and R-free are indeed unusually close in value. This, however, should not be a problem because a large gap between R-work and R-free usually indicates overfitting. In this particular case, as correctly guessed by the reviewer, the number of cycles of refinement was kept to a minimum towards the end of the refinement. This was done because additional refinement caused an important increase in R-free. We added this point in the methods section for clarity to the readers:

The R-work and R-free are unusually close in value. This, however, is not a problem because a large gap between R-work and R-free usually indicates overfitting. In this particular case, the number of cycles of refinement was kept to a minimum towards the end of the refinement. This was done because additional refinement caused an important increase in R-free.

- In the structure validation report, the clash score is high (23); typical values are below 5 for a medium resolution structure. The clashes listed in the table on page 9 should be inspected. This is also often improved by adding hydrogen atoms to the model and refining for many cycles.

We agree with the reviewer that the clash score is relatively high and that this can be improved through additional refinement. The problem is, however, that for this particular structure, it was not possible to decrease the clash score without significantly deteriorating the R-free. We did try to refine for many cycles, or for fewer cycles, we performed extensive manual rebuilding of the problematic regions (which are located near and at the cytoplasmic face, in particular some of the residues that are exposed to water and lipids of the lower leaflet). We also did reset B-factors and randomize coordinates multiple times over the course of refinement as we were worried about overfitting the data.

We did obtain a version of the model with a clash score of 5 by performing many cycles of refinement in Buster and removing the disulfide bond between CYS21 and CYS196, but all the other statistics were much worse. Visual inspection showed that this model was worse than the deposited model in terms of fitting the electron density, although there were no major differences between the two (except that the disulfide bond should be present). Here is a comparison of the overall quality that we obtained from the validation reports:

Many cycles of refinement versus deposited model:

R-free: 0.286 / 0.256

clash score: 5 / 23

Ramachandran outliers: 0.6% / 0.3%

sidechain outliers: 9.7% / 6.5%

RSRZ outliers: 6.6% / 6.0 %

In the end we decided to deposit the model that was most consistent with the x-ray data.

- The number of side chain outliers (6.5 %) is also high, with typical values around 1 %. The rotamer outliers in the table on page 13 should be inspected.

We did inspect all the rotamer outliers. Similarly to the clash score issue, the problematic residues are clustered in the part of the N-terminal region that is exposed to water, and on the bottom of TM1, TM2 and TM7, in particular some residues that are accessible to the

lipidic phase. The whole region has high B factors and relatively poor density for the side chains because of intrinsic disorder. For this reason, the conformations that best fit the electron density for these residues are not the best in terms of sidechain rotamers.

In addition, in our experience the structures refined with Buster tend to have worse sidechain rotamer statistics compared to, say, PHENIX. However we still like to use Buster because it seems to provide the most informative electron density maps for model building.

We do get better sidechain rotamer statistics if we refine our final model with Phenix, but again, the R-free gets much worse:

model refined in PHENIX versus deposited model:

R-free: 0.287 / 0.256

clash score: 15 / 23

Ramachandran outliers: 0.6% / 0.3%

sidechain outliers: 2.6% / 6.5%

RSRZ outliers: 3.7 % / 6.0 %

As mentioned earlier, our choice was to deposit the model that fitted best the X-ray data. We also tried to obtain the best Ramachandran plot we could, however we had to compromise with the clash score and side chain outliers given the flexibility observed in the bottom (cytoplasmic and lower membrane leaflet) part of the structure (see B-factors in Ext data Fig 1).

Reviewer #2 (Remarks to the Author):

The manuscript by Vasiliauskaitė-Brooks et al describes the first crystal structure of human alkaline ceramidase 3 (ACER3) to reveal a 7-TM architecture and active site similar to the adiponectin receptors. The structure provides insight into catalysis and a point mutation involved in disease. The manuscript is important and clearly written. There are some points that need to be clarified.

Major Points

1. There is concern with the accuracy of the modeled monoolein and whether it should be included in the refined structure given the ambiguity of the ligand identity. From the provided figure, it appears the polar section of monoolein resides in a highly hydrophobic environment, the electron density appears to extend beyond the polar headgroup section (ext Fig 3d), and there is a break in the electron density between the polar and hydrophobic sections near the Zn ion. Furthermore, the biological insight from modeling this is minimal.

We agree with all the points raised by the reviewer regarding the monoolein. The electron density for the ligand is relatively weak and somewhat discontinuous, and the chemistry is not ideal in terms of the environment of the polar head group. We want to emphasize that we tentatively modeled the density as a monoolein given its huge concentration in the LCP (54% by weight, almost 2M) and the accessibility of the pocket to the LCP, and we claim absolutely no biological insight from this ligand. We acknowledge that it would be standard practice to leave the ligand out given this ambiguity. We did however model it because we find it unsatisfactory to leave the pocket empty as this makes the structure physically unstable. In molecular dynamics simulations, we found that the pocket quickly collapses in the absence of a ligand and this leads to zinc unbinding.

We thus decided to leave the monoolein modeled but we clearly alert the reader that it is *tentatively* modeled as monoolein in the main text and in the figure legend.

2. Substrate specificity. The proposed steric hindrance mechanism for ceramide specificity versus sphingomyelin and glycosylceramide appears reasonable. However, the figure and text does not shown clearly what residues are blocking binding of larger lipids or how far the pocket extends below the ceramide headgroup.

We apologize for this lack of clarity. We have modified the figure by adding a new panel. The text in the legend was modified to clearly highlight the involved residues.

"(d) Left panel: close up view of the catalytic site with key residues shown as sticks and coloured as in (a). Right panel: W20, F80, H81 and D92 are shown as light blue spheres and are involved in the steric hindrance close to the primary alcohol of ceramide, supporting a possible mechanism for substrate selectivity."

3. The calcium and magnesium ions. Given Magnesium is present in the crystallization conditions, there should be some evidence or rationale included as to why the calcium ion is modeled as a calcium ion versus a magnesium ion. Secondly, where the magnesium ions bind in the crystal lattice is not shown, even though two are included in the refinement statistics.

The referee is right, we did not include the rationale for modeling a Ca^{2+} instead of a Mg^{2+} . This point was also raised by the two other reviewers. As indicated above, we did refine the structure with Mg^{2+} instead of Ca^{2+} and the calculated Fo-Fc maps clearly indicated that some electron were missing (positive signal in the Fo-Fc map). We apologize for not having included this information in the first version of the manuscript. To make this important point clear for the reader we are now showing the results of the Fo-Fc maps calculated either with Ca or with Mg in Extended data Figure 7. These difference maps, together with the average oxygen-metal distance of $\sim 2.3 \text{ \AA}$ and the observed coordination geometry, unambiguously indicate that the metal present in this site is indeed a calcium ion.

"Comparison of 2Fo-Fc and Fo-Fc maps calculated either with Ca²⁺ (a, b) and Mg²⁺ (c, d).

The positive signal indicated as a green mesh in (d) clearly indicates that the observed electron density cannot correspond to a Mg²⁺ ion."

In addition we have modified the Extended data Fig 1 to show where the Mg²⁺ and other ions were modeled in the structure.

Minor points

4. In extended data Fig. 2c, it is not clear what YFP-AdipoR2 and AdipoR2-YFP were co-expressed with. The assumption is ACER3-SNAP, but additional detail in the figure or figure legend would help clarify.

The figure was slightly modified to make this point clearer.

5. In Fig. 1B and 1C, please label the Ca ion for clarity. Also, the orientation to the membrane is lost in Fig. 1B, which would help identify the position of the Arg and Lys residues for putative anionic lipid binding.

The figure was modified to make this point clearer.

6. In fig. 2, can the electron density of the placed water molecule be shown?

Yes, it is a good point. We did add this feature on the new figure.

7. In Fig. 2, where is the intramembrane pocket accessible to the lipid leaflet and can you label these sites? In figure 3, it looks like the lower pocket (that accommodates the sphingosine moiety) is accessible, but is the upper pocket (where the acyl chain is proposed to bind) also accessible the membrane?

The fig 2 was modified to clarify all the points raised by the Reviewer #2.

The legend to Fig.2 was modified accordingly:

Figure 2 | ACER3 intramembrane domains. (a) View of the large hook-shaped internal cavity shown as surface (cavity mode 1) within the 7TM helix bundle (shown in light blue cartoon). The cavity is coloured according to the Eisenberg hydrophobicity classification. (b)(c) Close-up views of the pocket on the top highlighting the observed density (blue mesh, 2Fo-Fc map contoured at 1 σ) in which a water molecule was modeled (red sphere) (b) and on the side (c) with residues lining the pocket shown as sticks. (d) Close-up view of the zinc binding site highlighting the residues forming the first coordination sphere of the Zn²⁺ shown as sticks. The modeled water molecule is shown as a blue sphere. (e) 180° rotation of the view described in (a). (f) Side view of the pocket shown as surface (cavity mode 0) revealing the pocket accessibility at the level of the Zn²⁺ site and right above it.

8. Figure 3. Label sphingosine and acyl chains of ceramide for clarity.

We have added the labels as requested and modified slightly the figure and legends:

"View of the ceramide docking pose highlighting as sticks the residues in close proximity to the ceramide (coloured in cyan, and also shown alone on the right to indicate the identity of the fatty acid (FA) and sphingosine (SPH) chains)."

9. How will the modeled ceramide poses in ACER3 be made available to the scientific community? A pymol session?

The ceramide poses will be made available through a pymol session that will be made available to Nature Communications with the other manuscript files.

10. Should the title specify that this is alkaline ceramidase 3?

We will leave that decision to the editor.

Reviewer #3 (Remarks to the Author):

In this study, an X-ray crystal structure of human alkaline ceramidase type 3 (Acer3) at 2.7 Å resolution is reported. The enzyme is shown to consist of 7 transmembrane helices and to be a zinc hydrolase. Its regulation by calcium is explained on the basis of the crystal structure and a molecular rationale for the loss of function associated with mutants in progressive leukodystrophy is provided. The findings will be of interest to the membrane structural and functional biology communities as well as those interested in structure-based drug discovery and lipid signalling and metabolism. The work is done competently and the conclusions are reasonable and soundly based for the most part.

The following points should be addressed in a revision of the manuscript.

Describe the clinical phenotype of leukodystrophy and the ACER 3 E33G mutant in humans. Does a treatment exist for the disease?

We have modified the text in the introduction to mention the clinical phenotype as follows:

It was proposed that these aberrant levels of ceramides in the brain could result in an incorrect central myelination leading to the clinical phenotype associated with the ACER3 mutant, i.e. neurological regression at 6–13 months of age, truncal hypotonia, appendicular spasticity, dystonia, optic disc pallor, peripheral neuropathy and neurogenic bladder¹⁵.

Moreover we added a sentence to indicate that no treatment exists for the disease:

The critical role of ACERs in human physiology and, in particular ACER3, was recently revealed by clinical data demonstrating that ACER3 deficiency leads to progressive

leukodystrophy in early childhood¹⁵, a disease for which no treatment is available today.

"The molecular control of ACER enzymatic activity (agonists and antagonists) thus appears as a possible clinical intervention for the treatment of..."

This is an unsatisfactory statement. Advocating the possible use of antagonists and agonists in different diseases suggests that both types will have side effects that limit their utility. Accordingly, more research will be needed to show that modulation of ACER3 is a reasonable clinical intervention. The authors should rewrite this section placing less emphasis on clinical relevance and address the difficulties in developing drugs due to the broader implications associated with modulating ceramide homeostasis.

We apologize for placing too much emphasis on the clinical relevance of using antagonists or agonists in different diseases. We have rewritten the section to highlight the challenges that will need to be addressed to develop such drugs:

Modulating ceramide homeostasis can have broad implications and targeting ACER3 for clinical purpose will be extremely challenging. More research are needed to determine whether the molecular control of ACER enzymatic activity (agonists and antagonists) could constitute a possible clinical intervention for the treatment of leukodystrophy, colon cancer or acute myeloid leukemia among other pathologies involving ceramide dyshomeostasis. The first step towards this endeavour is to better understand the molecular basis of ACER function.

Explain how the lipid used in the crystallization trials amount to 54% by weight. The concentration of monoolein is at least a thousand times higher than reported. Show the revised calculations.

We apologize for this typo, where the "m" in mM was not removed during manuscript edition. We usually weigh out 10 mg of protein solution and 15 mg of lipids (a mix 10% cholesterol-90% monoolein). In the 15 mg of Lipids we have 13.5 mg of monoolein. This leaves with 13.5 mg of monoolein for 25 mg total ie 54% by weight.

The volume roughly equals 20 μ l after reconstitution. This gives \sim 13.5 mg/20 μ l of phase. 675 mg/ml \sim 1.9 M (molar range). We have corrected this typo in the main text.

Was a co-crystal structure of ACER3 with any sphingosine lipid attempted? This is an obvious thing to do and should be addressed in the manuscript.

We did attempt to co-crystallize ACER3 in the presence of sphingosine but so far these preparation did not yield any crystals. We are in the process of developing analogs of such compounds that would behave as irreversible ligands, since one issue might be that sphingosine affinity is not high enough to trap a stable conformation during crystallogenesis. This, we believe, is however beyond the scope of this manuscript. It could take, at best, several months of work before we obtained this structure, if at all.

The orientation of the monoolein in the putative binding pocket would appear to differ from the proposed substrate-binding pose. The authors should comment on this.

As now indicated the acyl chain moiety of monoolein is positioned in the same pocket as the acyl domain of ceramide in the proposed binding pose.

We have clarified this point in Extended data Fig 3 and in the legend.

"**(a)** Representation of the hook-shaped cavity within the ACER3 7TM *with the ceramide binding pose.* **(b) (c)** *Position of the modeled monoolein and representation of the*

calculated 2Fo-Fc map contoured at 1 σ (b) as well as the polder OMIT map (ref 21 in the main text) contoured at 2.85 σ (c)."

ED Fig. 5a. It is hard to distinguish covalent bonds from polar interactions. The difference should be obvious in a revised version of the figure.

We are sorry for the lack of clarity. The differences have been made obvious in the revised figure.

Did the authors perform MDS using ceramide containing the less preferred 18:0 acyl chain? It seems logical to perform this simulation to address the substrate specificity issues raised in the manuscript.

It would indeed be logical to perform this simulation. However, we did not attempt to address the issue of C18:1 vs C18:0 ceramide specificity using M.D. simulations yet.

In fact, we believe it would be worthwhile to fully address the substrate specificity, as well as thermodynamic and kinetic parameters of ACER3 activity combining experiments and simulations. Those are not straightforward and will require significant efforts that would necessitate a follow-up study.

-Firstly, the hook shaped pocket that accommodates the acyl chain should sterically favor the unsaturated lipid and restrict the conformational flexibility of C18:0. The resulting loss of entropy upon binding the protein should be greater for C18:0 than for C18:1. This probably will not manifest as any obvious changes in MDS if the simulations are started from the bound state. Instead, one would need to simulate the binding process starting from the unbound state.

-Secondly, the three polar side chains that are located in the vicinity of the C18:1 double bond (S99, Y149 and S228) are likely involved in stabilizing the interaction with the unsaturated acyl chain (interaction between pi electrons and the hydroxyls). Such interactions are not accurately described by the current force fields, and comparing the stability of C18:0 and C18:1 bound states would be best done using a polarizable force field which, as of today, do not include parameters for ceramides.

State how the identity of calcium was verified in the crystal structure and provide citations to the effect calcium has on ceramidase activity.

The referee is right, we did not include the rationale for modeling a Ca^{2+} instead of a Mg^{2+} . This point was also raised by the two other reviewers. As indicated above, we did perform structure refinement with both Mg^{2+} and Ca^{2+} over the course of solving the ACER3 structure. In the presence of Mg^{2+} the calculated Fo-Fc map clearly indicated that some electrons were missing (positive peak in the Fo-Fc map). We apologize for not having included this information in the first version of the manuscript. To make this important point clear for the reader we are now showing the results of the Fo-Fc maps calculated either with Ca^{2+} or with Mg^{2+} in Extended data Figure 7. These difference maps, together with the average oxygen-metal distance of $\sim 2.3 \text{ \AA}$ and the observed coordination geometry, unambiguously indicate that the metal present in this site is indeed a calcium ion.

The references for the Ca^{2+} were cited in the introduction and we forgot to include them again in the results section, we apologize for this mistake. We have now included those citations.

The authors describe and compare the different conformations of ADIPORs and ACERs, with a focus on TM4 and TM5. For the edification of the general reader, the process mediated by the conformational change should be explained. The authors refer to open and closed states. Does this mean the conformational change enables substrate binding and/or product release at ADIPORs? Would ACER3 be able to undergo a conformational change similar to that undergone by ADIPORs? The authors should address the issue of substrate access to the putative binding pocket in ACERs.

Those are all excellent questions that we are currently investigating by biophysical means. At this stage the conformational changes associated with intramembrane ceramidase activity is a hypothesis that we are currently exploring. These questions are not trivial and are difficult to address as biophysical analyses of membrane protein are technically challenging. In particular, we are developing the use of NMR and fluorescence spectroscopy as previously done for GPCR in my group (Sounier *et. al.* **Nature** 2015, 524, 375–378). The data supporting our hypothesis would take approximately two more years of work. We thus believe that they are beyond the scope of our manuscript.

We have added the following sentence to discuss this point with the reader:

"The three distinct conformations might however represent distinct steps of the common catalytic process, *the conformational changes and dynamics of such action of intramembrane ceramidases remains to be explored.*"

The loss of ceramidase activity in E33G ACER3 mutants might well originate earlier in the enzyme's path to maturity. Specifically, the E33G mutation is likely to eliminate calcium binding and early folding. The protein may never make it to its destination membrane. This issue should be addressed.

The referee is absolutely right. The E33G ACER3 mutant may encounter several problems

during early maturation. This is a key point that was addressed in the recently published study by Edvardson *et.al*. In fact, this study clearly demonstrated *in vivo* and *in vitro* (using both patients' cells and microsomes from yeast strains devoid of intrinsic ceramidase activity) that the mutant E33G leads to a complete loss of ceramidase activity despite similar expression level in the membrane fractions (i.e. the mutant E33G makes it to the membrane).

We added this important information in the manuscript to make this point clear to the reader:

"Remarkably, this discovery provides molecular insights into the E33G ACER3 mutation carried by patients suffering leukodystrophy, which results in the loss of ACER3 ceramidase activity despite similar level of expression than in control membrane preparations (15)."

Include a single pose of ceramide somewhere in Fig. 3a where its parts can be clearly seen and understood by the reader.

We have modified the Fig.3 to highlight where the FFA chains and the sphingosine moiety of the ceramide sit in the model.

Reviewers' Comments:

Reviewer #1:

Remarks to the Author:

Most of the minor points have been addressed by the authors. However, in the legend of Extended Data Figure 4, it should be specified that the asterisks indicate metal-binding residues.

Regarding the mutational analysis, I wasn't asking for the development of new assay, but simply to repeat the activity assay of Extended Fig. 1 on selected mutants of the active site that can in fact be purified (i.e. not mutants that would affect Ca-binding and therefore the structural integrity).

Also, I am concerned whether the structure has been refined to convergence. The authors indicate that the number of refinement cycles were minimized. This could artificially make the Rfree and R-factor close to one another at the possible expense of structure geometry. At 2.7 Å resolution, some overfitting is a normal part of the refinement process. Typically, at this resolution one would expect a difference of about 5 to 10% between R and Rfree.

Also concerning is that the authors notice that additional refinement increases the Rfree. By how much does it increase? This should not normally happen and I am wondering if this is due to the data processing procedure used here or some other systematic issue with the data?

Although Rmerge is generally not a good statistic for determining data cutoff resolution, in this case, it is rather high even in the low resolution bins (12% at 12 Å is high) where the data should be strong and agree well. Although this is likely due to the procedure used here of collecting and combining small wedges of data from many crystals, I am wondering if the rejection cut-off value for dataset inclusion is not stringent enough?

In any case, the refinement should be run to convergence such that both R-free and R-factor no longer change significantly.

Finally, I have concerns on the rather large discrepancy between the Wilson B (35.3) and the model B-factors which are quite a bit higher, particularly for the metals. The protein B-factors are also quite high (~70). Although the Wilson B is determined based on assumptions and will not match the average model B-factors exactly, they should at least be in the same ballpark.

Are the metal B-factors close in value to the atoms to which they are bound? Was the data very anisotropic? If anisotropy is present, this could be addressed by using ellipsoidal truncation. As a validation, the authors should calculate intensities from the model and confirm that the B-factor from them is close in value to the observed Wilson B.

Reviewer #2:

Remarks to the Author:

All points addressed satisfactorily.

Reviewer #3:

Remarks to the Author:

The authors have responded adequately to most of this reviewer's comments. However, a number of issues remain that require clarification and response.

Grammatical and compositional errors, especially those that appeared in the revised manuscript,

will need correcting.

163. Why is a hydrophilic patch (assuming it is a constriction of some kind) needed to block an apolar chain? An apolar block would do exactly the same thing. This should be expanded on in the manuscript.

169. Include in the manuscript the calculated molar concentration of monoolein in the mesophase.

480. For clarity, report the exact composition of the inner and outer layers in each situation modelled.

487. Is this an appropriate reference for lipid composition?

443 and Table 1. It should be clearly stated in the manuscript that these are two sets of serial data collected from many crystals (set 1, 77; set 2, 198). With serial data of this type, indicators like Rmerge and Rmeas are not suitable for assessing data quality because a few weak data sets with high Rmerge will throw off the statistics. In such cases, indicators like CC1/2 and I/sig are more appropriate. The values reported in Table 1 (CC1/2 ~50% and I/sig ~1 in the highest resolution shell) are reasonable.

Table 1. A refined protein B-factor value of ~70 is reasonable. However, a Wilson B of 35 is too low. Processing statistics should be reanalysed.

All figures. Label transmembrane helices for clarity.

Figure 2. Provide an explanation in the legend for the hydrophobicity scale used.

It is hard to tell from Figure 2 what parts of the cavity are open to the membrane. The problem may have to do with the use of transparent helices. This makes it difficult to tell what is in front of or behind the cavity.

ED Figure 3. Would it not make more sense to model the monoolein with its polar headgroup closer to the zinc and catalytic residues and its tail extending to fill the pocket? A second monoolein might be accommodated in the resulting empty side pocket with the methyl end of its acyl chain extending out into the membrane where it is disordered and no longer in density. These are suggestions. It is hard for this reviewer to judge by looking at 2D images.

Small quantities of diolein can be present in LCP samples. It is possible that diolein is in the pocket. This possibility should be considered.

'Was a co-crystal structure of ACER3 with any sphingosine lipid attempted? This is an obvious thing to do and should be addressed in the manuscript.' Please comment on this IN THE MANUSCRIPT.

We would like to thank again the referees for the constructive comments. The manuscript has been revised to address their main concerns and comments. Below, we provide a point-by-point response. Each referee's comments are in coloured fonts and our response is in normal black font. Text changes made in the manuscript are in italic underlined fonts.

Reviewer #1 (Remarks to the Author):

Most of the minor points have been addressed by the authors. However, in the legend of Extended Data Figure 4, it should be specified that the asterisks indicate metal-binding residues.

This information was added in the legend.

Regarding the mutational analysis, I wasn't asking for the development of new assay, but simply to repeat the activity assay of Extended Fig. 1 on selected mutants of the active site that can in fact be purified (i.e. not mutants that would affect Ca-binding and therefore the structural integrity).

Originally, this reviewer asked for some additional functional assays of several mutants including S99, Y149 and S228 to validate the docking and most importantly other Ca binding residues in addition to E33. We thus have performed the requested experiments, implicating the mutagenesis, expression, production and purification of seven mutants compared to ACER3-BRIL : S99A, Y149A and S228A and D19G, E22G, N24G and E33G. Details regarding this study are now indicated in the method section.

"Mutants were generated by site-directed mutagenesis, confirmed by DNA sequencing (MWG-Eurofins) and expressed and purified as described for the ACER3-BRIL construct.

Ceramidase activity assays were performed by incubating purified ACER3-BRIL wild-type or mutants (1 μ M) with ceramide (20 μ M)....."

The presented data for enzymatic activity are representative of three experiments performed on three independent ACER3-BRIL enzyme preparations and two independent mutants preparations, each experiment contained six replicates.

-The data for the S99A, Y149A and S228A are shown in EDF5. They show that the Y149A mutant significantly impaired the enzymatic activity of ACER3 while the serine mutants do not present any functional differences with the ACER3-BRIL Wt enzyme. None of the mutants showed a change in the substrate preference of ACER3, suggesting that they alone are not critically involved in this process. We have reworded this hypothesis in our manuscript to take into consideration these functional data:

We functionally tested S99A, Y149A and S228A mutants and compared their enzymatic activity with the one of ACER3-BRIL wild-type preparations (WT). In agreement with the docking pose, the Y149A mutant presented an important decrease in activity, while S99A and S228A mutants did not show any significant functional differences (Extended Data Fig. 5).

Moreover, none of the mutants showed a change in the substrate preference (Extended Data Fig. 5), suggesting that they alone are not critically involved in this selectivity.

The data for the D19G, E22G, N24G and E33G are shown in EDF7. As anticipated, we confirmed the functional data already published for the E33G mutant, i.e. a dramatic decrease in the enzymatic function when compared to WT. Two other mutants E22G and N24G behaved as E33G with an important decrease of enzymatic activity while, surprisingly, the enzymatic activity of D19G was only partially affected. We added the description of this data in our revised manuscript:

In order to further validate the role of the Ca²⁺ binding site, we performed some additional enzymatic assays on ACER3 single point mutants D19G, E22G, N24G and E33G. As anticipated, we confirmed the functional data already published for the E33G mutant i.e., a dramatic decrease in the enzymatic function when compared to the wild-type preparations (Extended data Fig. 7). Two other mutants, E22G and N24G, behaved as E33G, displaying a clear decrease in enzymatic activities (Extended data Fig. 7). Surprisingly, the enzymatic activity of D19G was only partially affected (Extended data Fig. 7). Altogether, these data confirm the critical role of the Ca²⁺ binding site in ACER3 function.

The legends of extended figures have also been modified:

EDF5:

(c) Close up view of the S99, Y149, S228 domain (left panel) and enzymatic assays performed with the C18:1 (black bars) or C18 (grey bars) substrates showing the Area Under the Curve (AUC) sphingosine signal normalized to the internal standard (Sph d17:1) for ACER3-BRIL, ACER3-BRIL-S99A, ACER3-BRIL-Y149A and ACER3-BRIL-S228A mutants (right panel). The results shown are the mean ± s.d. of two independent experiments

EDF7:

(h) Enzymatic assays performed with the C18:1 substrate showing the Area Under the Curve (AUC) sphingosine signal normalized to the internal standard (Sph d17:1) for ACER3-BRIL, ACER3-BRIL-D19G, ACER3-BRIL-E22G, ACER3-BRIL-N24G and ACER3-BRIL-E33G mutants. The results shown are the mean ± s.d. of two independent experiments performed in pentaplicate.

Altogether, the performed biochemical/functional study is now supporting the hypotheses derived from the structure, in particular for the critical role of the Ca²⁺ binding site.

Also, I am concerned whether the structure has been refined to convergence. The authors indicate that the number of refinement cycles were minimized. This could artificially make the R_{free} and R-factor close to one another at the possible expense of structure geometry. At 2.7 Å resolution, some overfitting is a normal part of the

refinement process. Typically, at this resolution one would expect a difference of about 5 to 10% between R and R_{free}.

Also concerning is that the authors notice that additional refinement increases the R_{free}. By how much does it increase? This should not normally happen and I am wondering if this is due to the data processing procedure used here or some other systematic issue with the data?

As requested by the reviewer, we have performed additional cycles of refinement and (very minor) manual changes in coot. The number of refinement cycles in Buster was increased to enable the refinement to converge, which decreased the clashscore from 23 to 8 – a more reasonable value at this resolution. R_{free} increases to about 27% upon additional refinement. We agree with the reviewer that this should not normally happen, and as pointed out by the reviewer it is likely due to the data processing procedure. However, the R_{free} values reported are within an acceptable range for this resolution and small inaccuracies in data processing are quite common when merging a large number of data sets from crystals that are not exactly identical.

Although R_{merge} is generally not a good statistic for determining data cutoff resolution, in this case, it is rather high even in the low resolution bins (12% at 12 Å is high) where the data should be strong and agree well. Although this is likely due to the procedure used here of collecting and combining small wedges of data from many crystals, I am wondering if the rejection cut-off value for dataset inclusion is not stringent enough?

We used CC1/2 and I/σ parameters to decide on data merging and resolution cutoffs, which is standard procedure, in particular for SSX data. As mentioned by the reviewer, R_{merge} is not a good statistic to determine data cutoff resolution because low multiplicity or completeness for a few weak dataset in the low resolution shell will result in high R_{merge} values. Unfortunately there is no golden rule for deciding where to cut the data and the current data set was good enough to solve the structure and obtain reasonable statistics.

In any case, the refinement should be run to convergence such that both R-free and R-factor no longer change significantly.

We have run the refinement to convergence as requested. R-factor/R-free are stable at 24.9/27.1 %. These values were inserted in the new Extended data Table 1. We also modified the deposited data accordingly. A new validation report is also made available for the manuscript revision.

Finally, I have concerns on the rather large discrepancy between the Wilson B (35.3) and the model B-factors which are quite a bit higher, particularly for the metals. The protein B-factors are also quite high (~70). Although the Wilson B is determined based on assumptions and will not match the average model B-factors exactly, they should at least be in the same ballpark.

We investigated this issue and found that the low Wilson B value of 35.3 for the native data sets comes from an incorrect calculation by the program CTruncate. We now report the average Wilson B from the individual 10° data sets that went into merging which is 73.6. We apologize for this mistake and thank again the reviewer for notifying this point.

Are metal B-factors close in value to the atoms to which they are bound? Was the data very anisotropic? If anisotropy is present, this could be addressed by using ellipsoidal truncation. As a validation, the authors should calculate intensities from the model and confirm that the B-factor from them is close in value to the observed Wilson B.

The zinc and calcium ions have high B factors because they are located close to the cytoplasmic side of the protein. As can be seen in Extended Data Figure 1 d and e, there is a gradient of B factors going from C- to N-terminus. The protein atoms that are bound to the metals have similar B values. The data was only moderately anisotropic (see aimless.log below).

Estimates of resolution limits in reciprocal lattice directions:

Along h axis

from half-dataset correlation $CC(1/2) > 0.30$: limit = 2.95A

from $Mn(I/sd) > 1.50$: limit = 3.09A

Along k axis

from half-dataset correlation $CC(1/2) > 0.30$: limit = 2.70A == maximum resolution

from $Mn(I/sd) > 1.50$: limit = 2.79A

Along l axis

from half-dataset correlation $CC(1/2) > 0.30$: limit = 2.70A == maximum resolution

from $Mn(I/sd) > 1.50$: limit = 2.70A == maximum resolution

Reviewer #2 (Remarks to the Author):

All points addressed satisfactorily.

Reviewer #3 (Remarks to the Author):

The authors have responded adequately to most of this reviewer's comments. However, a number of issues remain that require clarification and response.

Grammatical and compositional errors, especially those that appeared in the revised manuscript, will need correcting.

163. Why is a hydrophilic patch (assuming it is a constriction of some kind) needed to block an apolar chain? An apolar block would do exactly the same thing. This should be expanded on in the manuscript.

We did not say that the hydrophilic patch is needed to block an apolar chain. We are sorry for this lack of clarity. We simply mention that the presence of the patch at this specific site in the structure may play a role in the binding of ceramide. The referee is right that apolar chains could also play a similar role. Those apolar chains are however not present in the structure. That is why we prefer not to discuss this point further.

169. Include in the manuscript the calculated molar concentration of monoolein in the mesophase.

This has been done (*i.e.* $\sim 1.9 M$).

480. For clarity, report the exact composition of the inner and outer layers in each situation modelled.

This has been done.

487. Is this an appropriate reference for lipid composition?

An appropriate reference was added to the text : (EMBO Rep. 2017 Nov;18(11):1905-1921. doi: 10.15252/embr.201643426.)

443 and Table 1. It should be clearly stated in the manuscript that these are two sets of serial data collected from many crystals (set 1, 77; set 2, 198). With serial data of this type, indicators like Rmerge and Rmeas are not suitable for assessing data quality because a few weak data sets with high Rmerge will throw off the statistics. In such cases, indicators like CC1/2 and I/sig are more appropriate. The values reported in Table 1 (CC1/2 $\sim 50\%$ and I/sig ~ 1 in the highest resolution shell) are reasonable.

We added a note to Extended Data Table 1 to clearly state that the two datasets correspond to serial data collected from many crystals. We also agree with reviewer about the comment on Rmeas or Rmerge. We have now added a note to Extended Data Table 1 and justified the argument with citation of Karplus and Diederichs 2015, Curr. Op. Biol., where data quality metrics for serial crystallographic data were discussed in details.

Table 1. A refined protein B-factor value of ~ 70 is reasonable. However, a Wilson B of 35 is too low. Processing statistics should be re-analyzed.

Low Wilson-B was due to a miscalculation done in CTruncate program. We now

report the average Wilson B from the individual 10° data sets that went into merging which is 73.6. We thank reviewer for drawing our attention to this issue. The corrected Wilson-B value has now been integrated in Extended Data Table 1.

All figures. Label transmembrane helices for clarity.

This was done.

Figure 2. Provide an explanation in the legend for the hydrophobicity scale used.

This was done. The following text was added:

"The cavity is coloured according to the Eisenberg hydrophobicity classification *from red (high hydrophobicity) to white (low hydrophobicity)*."

It is hard to tell from Figure 2 what parts of the cavity are open to the membrane. The problem may have to do with the use of transparent helices. This makes it difficult to tell what is in front of or behind the cavity.

We have modified the Figure 2 and its legend to clarify this point. The following text was added:

" the TM4 has been removed for clarity".

ED Figure 3. Would it not make more sense to model the monoolein with its polar headgroup closer to the zinc and catalytic residues and its tail extending to fill the pocket? A second monoolein might be accommodated in the resulting empty side pocket with the methyl end of its acyl chain extending out into the membrane where it is disordered and no longer in density. These are suggestions. It is hard for this reviewer to judge by looking at 2D images.

We have analyzed this possibility but unfortunately this would not make more sense.

Small quantities of diolein can be present in LCP samples. It is possible that diolein is in the pocket. This possibility should be considered.

The referee made a good point but it is very unlikely. We purchase highly pure monoolein (over 99%) and even if the 1% of impurity was diolein this would be a very low concentration of this compound. Anyway, diolein does not fit in the electron density.

'Was a co-crystal structure of ACER3 with any sphingosine lipid attempted? This is an obvious thing to do and should be addressed in the manuscript.' Please comment on this IN THE MANUSCRIPT.

As indicated in our first reply, we did attempt to co-crystallize ACER3 in the presence of sphingosine but so far these preparations did not yield any crystals. We feel that

this information is not critical and we could not find any logical ways to include such a negative data in the present manuscript.

Reviewers' Comments:

Reviewer #1:

Remarks to the Author:

The authors have addressed all of my comments. The new mutational data nicely supports the importance of Ca-binding residues and clarifies substrate preference. The refinement statistics now appear to be sound.

The one final edit I recommend is removing line 190 "It is not clear from the structure alone whether this feature has a role in the C18:1 vs C18:0 ceramide substrate preference.", since now the new mutational data does appear to clarify that these residues are not involved in substrate preference as stated later on line 197.

Reviewer #3:

Remarks to the Author:

My comments have been addressed in this revision.

We would like to thank again the referees for the constructive comments. The manuscript has been revised to address their comments. Below, we provide a point-by-point response. Each referee's comments are in coloured fonts and our response is in normal black font.

REVIEWERS' COMMENTS:

Reviewer #1 (Remarks to the Author):

The authors have addressed all of my comments. The new mutational data nicely supports the importance of Ca-binding residues and clarifies substrate preference. The refinement statistics now appear to be sound.

The one final edit I recommend is removing line 190 "It is not clear from the structure alone whether this feature has a role in the C18:1 vs C18:0 ceramide substrate preference.", since now the new mutational data does appear to clarify that these residues are not involved in substrate preference as stated later on line 197.

This was done.

Reviewer #3 (Remarks to the Author):

My comments have been addressed in this revision.